# Whole-genome sequencing analysis identifies rare, large-effect noncoding variants and regulatory regions associated with circulating protein levels

Gareth Hawkes [1,3] ✉, Kartik Chundru [1], Leigh Jackson [1], Kashyap A. Patel [1], Anna Murray [1], Andrew R. Wood [1], Caroline F. Wright [1], Michael N. Weedon [1,3] ✉, Timothy M. Frayling [1,2,3] ✉ & Robin N. Beaumont [1,3] ✉

The contribution of rare noncoding genetic variation to common phenotypes is largely unknown, as a result of a historical lack of population-scale whole-genome sequencing data and the difficulty of categorizing noncoding variants into functionally similar groups. To begin addressing these challenges, we performed a *cis* association analysis using whole-genome sequencing data, consisting of 1.1 billion variants, 123 million noncoding aggregate-based tests and 2,907 circulating protein levels in ~50,000 UK Biobank participants. We identified 604 independent rare noncoding single-variant associations with circulating protein levels. Unlike protein-coding variation, rare noncoding genetic variation was almost as likely to increase or decrease protein levels. Rare noncoding aggregate testing identified 357 conditionally independent associated regions. Of these, 74 (21%) were not detectable by single-variant testing alone. Our findings have important implications for the identification, and role, of rare noncoding genetic variation associated with common human phenotypes, including the importance of testing aggregates of noncoding variants.

Rare genetic variants in noncoding regions of the human genome can cause severe rare disease[1,2], but their role in common, complex traits is still largely unknown. Array-based imputation, genome-wide association studies (GWASs) have identified tens of thousands of common variant associations with human disease[3], most of which are outside coding regions[4]. However, efforts to identify rare variants associated with common phenotypes have been largely limited to the coding regions using exome-sequencing data, with a mixture of single-variant testing (low statistical power but clearer functional interpretation)

and aggregate testing (higher statistical power but can be difficult to interpret). Our ability to assess rare variants in the noncoding genome has until recently been limited, because of the lack of whole-genome sequencing (WGS) data in large studies and the difficulty of defining biologically meaningful noncoding regulatory genomic units.

The identification of noncoding regulatory elements using WGS data could provide important insight into gene regulation that complements knowledge gained from exome sequencing and array-based studies. Based on population genetic metrics such as constraint, the

[1]Department of Clinical and Biomedical Sciences, Faculty of Health and Life Sciences, University of Exeter, Exeter, UK. [2]Faculty of Medicine, Department of Genetic Medicine and Development, CMU, Geneva, Switzerland. [3]These authors contributed equally: Gareth Hawkes, Michael N. Weedon, Timothy M. Frayling, Robin N. Beaumont. ✉e-mail: g.hawkes2@exeter.ac.uk; m.n.weedon@exeter.ac.uk; t.m.frayling@exeter.ac.uk; r.beaumont@exeter.ac.uk

amount of functional noncoding DNA has been estimated to be 4–5× greater than the amount of coding sequence[5] and 10% and 6% of promoters and enhancers, respectively, are under as much mutational constraint as coding regions[6]. WGS allows us to examine the role of intronic, proximal and distal regulatory elements and covers the entire allele frequency spectrum in a population, including a large proportion of variants that are observed only once or twice even in a very large sample.

There are very few studies of WGS data in the context of common phenotypes. Recent examples from TOPMed[7] considered lipid levels[8,9] ($N$ = 66,000) and blood pressure[10] ($N$ = 51,456) and a study from UK Biobank (UKB) on standing height[11] ($N$ discovery = 200,003): these studies found a limited number of new signals, possibly because of the relatively small sample sizes for the detection of new rare variants. WGS has also been used for identification of genetic variation associated with protein levels in studies of up to 3,000 individuals[12–14].

The UKB's release of circulating protein data, in combination with WGS, provides an unprecedented opportunity to test the impact of rare noncoding genetic variation on common, biologically proximal and well-measured human phenotypes[15]. Three recently published studies[15–17] described the 2023 release of these data on 2,923 circulating proteins in 54,306 individuals. First, Eldjarn et al.[15] identified 30,062 protein quantitative trait loci (pQTLs) in single-variant analysis of genetic data imputed from 150,119 WGS analyses from UKB with 2,931 measured protein levels and compared results with proteomics derived from an Icelandic WGS cohort with whole-genome sequences. Second, Dhindsa et al.[16] identified 5,433 pQTLs in an exome-sequencing analysis and performed aggregate testing within coding regions, identifying 1,962 gene–protein associations. Finally, Sun et al.[17] identified 14,287 pQTL single variants using a conventional imputation-based GWAS.

Using WGS data and circulating protein levels as exemplar traits, we tested two related hypotheses: (1) rare noncoding single genetic variants, not currently detectable by imputation or exome sequencing, contribute to common human phenotypes with similar effects to coding variants and (2) aggregates (groups) of rare noncoding genetic variants in regulatory regions of the genome are associated with human phenotypes, analogous to gene-level burden analyses in coding sequences. Importantly, and in contrast to the previous three papers on the UKB proteomic data, we used the full range allele frequencies of DNA sequence variation detected with short-read WGS for all participants with circulating protein measurements, providing information on ~1.1 billion variants, but limited the search for each phenotype to *cis*-regions around the protein-coding gene from which the protein derived.

We performed primary discovery association analyses for 2,903 measured circulating protein levels using annotated WGS data on 46,362 individuals of inferred European genetic ancestry from the UKB, a population cohort from the United Kingdom. In addition, we performed single-variant analysis in 899 and 1,098 individuals of inferred south Asian and African ancestry, respectively. Our analyses identified 1,651 high-quality single variants, contrasted the impact of coding and noncoding variants, highlighted the power of rare variant noncoding aggregate testing by identifying 357 independent loci and demonstrated the importance of accounting for coverage in WGS-based association studies.

## Results

Twelve proteins that were either fusion proteins or did not directly match to an *HGNC* gene symbol were excluded (Supplementary Table 1). For each measured protein, we performed both single-variant (minor allele count (MAC) ≥ 5) and genomic aggregate association tests (minor allele frequency (MAF) < 0.1%) in a *cis*-window around the gene coding for the protein, extending 1 Mb from the 5′- and 3′-UTRs, based on the most extreme 5′- and 3′-ends of any GENCODE transcript of the gene. We used 1 Mb as the approximate distance recently identified as the boundary between when a pQTL is more likely to be a *cis* rather than

a *trans* association[18]. Circulating protein level measurements were rank-inverse normalized at runtime, and age, age[2], sex, recruitment center, WGS center, time since blood draw, fasting time, 40 genetic principal components (PCs) and Olink batch ID were included as covariates (Methods). In total, despite only considering a 2-Mbp *cis*-window per protein, we tested 128,434,590 single (including single-nucleotide variants and small insertions or deletions (indels)) and structural variant associations with MAC ≥ 5. We identified independent variant associations using a modified version of GCTA-CoJo[19] (Methods).

We annotated all genetic variants using Ensembl's Variant Effect Predictor v.110.1 (VEP)[20] (Methods) and used the output to categorize variants as gene centric (for example, coding, predicted intronic splicing, intronic unspecified and proximal regulatory) and intergenic regulatory (for example, Ensembl regulatory regions, noncoding RNA and intergenic unspecified) for aggregate-based association testing. Noncoding variants were annotated as gene centric if they were within either the UTRs themselves or the intronic regions of the gene, or within 5 kbp of the UTRs. In addition, we performed aggregate testing on all noncoding variants in overlapping (1-kb overlap) 2-kb sliding windows. We additionally subcategorized variants within a subset of aggregate units by measures of constraint (JARVIS[21]), conservation (genomic evolutionary rate profiling (GERP)[22]) and/or predicted deleteriousness (combined annotation-dependent deletion score (CADD)[23]). To identify independent, rare, noncoding aggregate associations, we adjusted noncoding aggregate tests for common pQTLs (MAF > 0.01) and all variants annotated as coding within the gene coding for the protein itself, henceforth referred to as the cognate gene, regardless of variant frequency (Fig. 1). In total, we performed association testing of 123,598,575 aggregates including 182,136,116 variants with an MAF < 0.1%, up to and including singletons. Statistical significance was defined based on simulation studies (Methods): $P < 2.95 \times 10^{-10}$ for single variants and $P < 8.71 \times 10^{-9}$ for genomic aggregates.

### We identified 13,457 candidate pQTLs across 2,891 proteins

For the 2,891 circulating proteins, we identified 13,457 statistically independent *cis*-pQTL associations (MAC ≥ 5), 293 of which were structural variants (Figs. 2 and 3 and Supplementary Table 2). We identified at least one *cis* association for 2,036 proteins, with a median of 5 independent pQTLs per circulating protein; 429 proteins were associated with >10 independent pQTLs, including a maximum of 49 for CD177. These results are consistent with a previous study using 150,000 UKB genomes and imputed into the remaining samples using the same data[15].

### Over half of independent pQTLs may be coverage artefacts

As our study aimed to assess the role of noncoding variants on protein levels, we jointly modeled all genome-wide significant variants, coding and noncoding (Methods), to avoid confounding by haplotype effects. However, we subsequently found that proteins with the largest number of statistically independent pQTLs were substantially enriched for regions of low coverage over the cognate gene (Supplementary Tables 2 and 3). For example, CD177 (MANE Select transcript ENST00000618265.5) with 49 independent pQTLs has little-to-no coverage over exon 5 or intron 6 (Extended Data Fig. 1). There was also an enrichment for proteins where expression of the cognate gene was largely driven by copy number of repeat polymorphisms. For example, FCGR3B[24] (43 independent signals), AMY2A (25 independent signals) and AMY2B (27 independent signals) are encoded by genes known to have highly variable copy numbers, with amylase levels largely determined by the number of copies of the latter two genes. Furthermore, lipoprotein (a) levels (34 independent signals) are largely explained by Kringle's repeat polymorphisms[25]. For each of these genes, the most likely explanation for the large number of independent signals is that true causal variant(s) have not been adequately adjusted for, giving rise to artificially high numbers of independent pQTLs. We have shown this previously for expression QTLs[26,27].

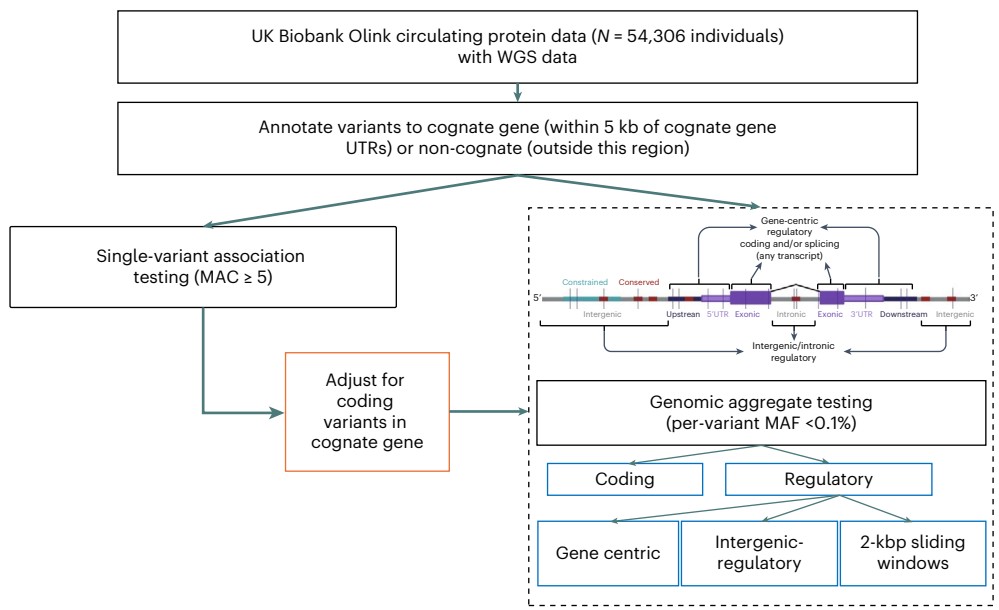

**Fig. 1 | Study design for pQTL single and aggregate genetic variant analysis.** Flow chart demonstrating the study design.

To help ensure that we include only genuinely independent causal variants, we limited subsequent analyses to genes well covered over the locus (n excluded = 1,247; Supplementary Table 3 and Methods). We also excluded proteins with a substantial difference in regression coefficients before and after joint conditional analysis, because this can be indicative of an uncalled causal variant[26] (n excluded = 618; Supplementary Table 4). Subsequent analysis revealed that these low-coverage regions were most strongly enriched (Methods) for genomic regions previously reported[28,29] to contain segmental duplications (odds ratio (OR) = 1.10 (1.07, 1.12), $P = 1.46 \times 10^{-17}$ per overlapping segmental duplication; Supplementary Table 3), suggesting that these low-coverage regions are not unique to the UKB WGS dataset. After these filtering steps, we retained 1,026 proteins for further analysis.

### We identified 1,651 high-quality rare pQTLs for 599 proteins

After these filters, we were left with 5,076 cis-pQTLs, including 98 structural variants (Supplementary Table 5). The mean variance (within sample) explained jointly by all independent cis-pQTLs for a given protein was 4.10% (median 2.02%), similar to estimates previously reported in ref. 17. Of the 5,076 variant associations identified, 762 (15.0%) were in the rare frequency range (0.1% ≤ MAF < 1%) and 925 (18.2%) were very rare (MAF ≤ 0.1%), including 36 structural variants. We refer to the 1,651 single-nucleotide or short indel variants in both frequency bins as rare pQTLs hereafter.

We additionally performed single-variant testing for 1,184 and 1,027 individuals of genetically inferred south Asian and African ancestry, respectively, with both WGS and Olink proteomic data. Across the three genetically inferred ancestries considered (European (EUR), south Asian (SAS) and African (AFR)), we observed a strong correlation of effect sizes for pQTLs between EUR and SAS individuals (r = 0.797, $P < 1 \times 10^{-300}$) and weaker correlation between EUR and AFR individuals (0.656, $P < 1.00 \times 10^{-300}$) and AFR and SAS individuals (0.527, $P = 6.79 \times 10^{-250}$). Despite the much smaller sample sizes for the SAS and AFR analyses, we identified 228 and 396 independent pQTLs, respectively (Supplementary Table 6). Of these, 77 (33.8%) and 70 (17.7%) were also the lead pQTL in our analysis of EUR individuals.

We compared our single-variant pQTL results with those of Eldjarn et al.[15], who analyzed genomic data imputed from a release of 150,119 UKB WGS analyses in the 54,306 individuals with proteomic data. We found 2,575 of their 3,386 cis-pQTLs (within the regions we tested; 76.1%), which directly mapped to the UKB DRAGEN WGS calls, were in strong linkage disequilibrium (LD) ($r^2 \geq 0.8$) with at least one of our signals for the same circulating protein. The overlap was larger (848 out of 904; 93.8%) when considering only rare variants (MAF < 1%). On average, we identified 5.0 (median 4) independent cis-pQTLs (rare and common) per protein, compared with Eldjarn et al.[15] who identified 10.1 (median 8).

### Most coding pQTLs reduce circulating cognate protein levels

As expected, and consistent with the recent exome-sequencing study[16], 96.3% of pQTLs annotated as high-confidence loss of function, as defined by LOFTEE[30], were associated with reduced circulating protein levels (Supplementary Information, Supplementary Table 7 and Extended Data Fig. 2).

Aggregate-based testing of rare coding variants identified 545 genes associated with 539 circulating protein levels after adjusting for common (>0.1%) pQTLs from our single-variant analysis (Supplementary Table 8). Of these genes, 14 were not identified by the exome-sequencing analyses performed by Dhindsa et al.[16] (Supplementary Table 9), potentially as a result of differences in sequencing coverage, demonstrating WGS's ability to discover coding variation not captured by exome sequencing.

### Noncoding pQTLs are enriched in 5′-UTRs

We identified 604 independent rare noncoding single-variant–protein associations with 369 proteins (Figs. 2 and 3 and Supplementary Table 10), compared with rare coding variation (985 high-confidence loss-of-function or missense variants). We stratified pQTLs into cognate and noncognate groups based on annotation and the most severe consequence[31], with priority given to annotations relative to the cognate gene. Of the 604 noncoding pQTLs, 343 (56.8%) were annotated as regulatory for the cognate gene (that is, within 5 kbp of the UTRs or intronic; Fig. 3d). This contrasts with 972 (98.7%) rare coding variants annotated to the cognate gene.

Previous studies comparing two different platforms have suggested a high proportion of cis-pQTLs could be caused by epitope-binding artefacts. Although we did not have access to SomaLogic data that would enable a direct comparison with Olink, we noted that none of our 604 rare noncoding variants was in at least 'low' LD ($r^2 \geq 0.1$) with a coding variant. We also found no evidence of

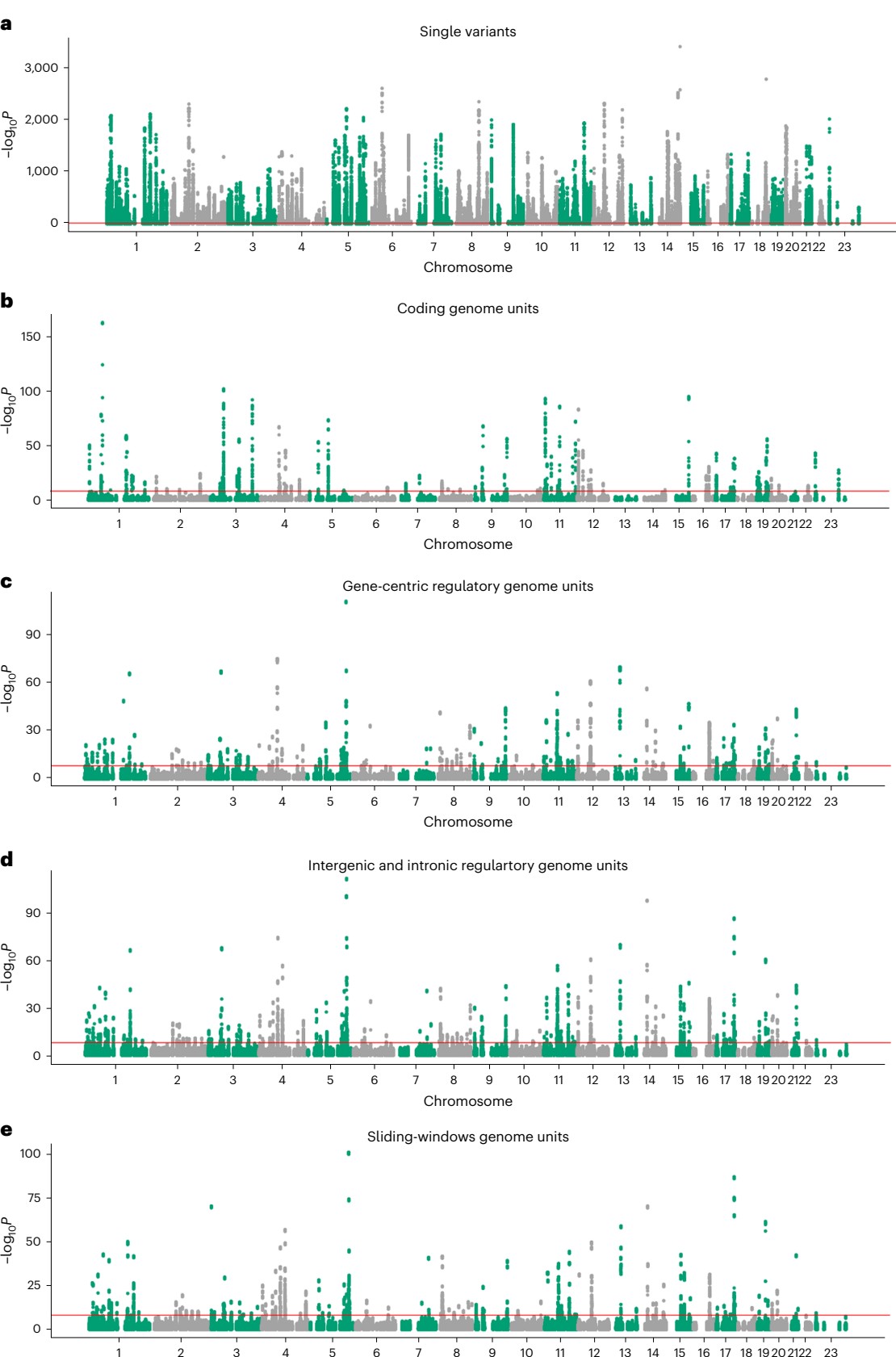

**Fig. 2 | Manhattan results of single-variant and aggregate pQTL analyses.**
Manhattan plots showing associations between *cis* variants and regions with
circulating protein levels, after adjusting for associated common variants and
all coding variants of the cognate gene. **a–e**, The *x* axis represents genomic
position and the *y* axis shows $-\log_{10}P_{\text{two-sided}}$ for our *cis* results across all proteins,
split into single variants (**a**), coding aggregates (**b**), gene-centric regulatory
(proximal) aggregates (**c**), intergenic and intronic regulatory aggregates (**d**) and
sliding-window aggregates (**e**). The red lines represent Bonferroni's significance
thresholds ($P \leq 2.95 \times 10^{-10}$ for single variants and $P \leq 8.71 \times 10^{-9}$ for aggregate
tests). *P* values are derived from two-sided tests from mixed linear models.

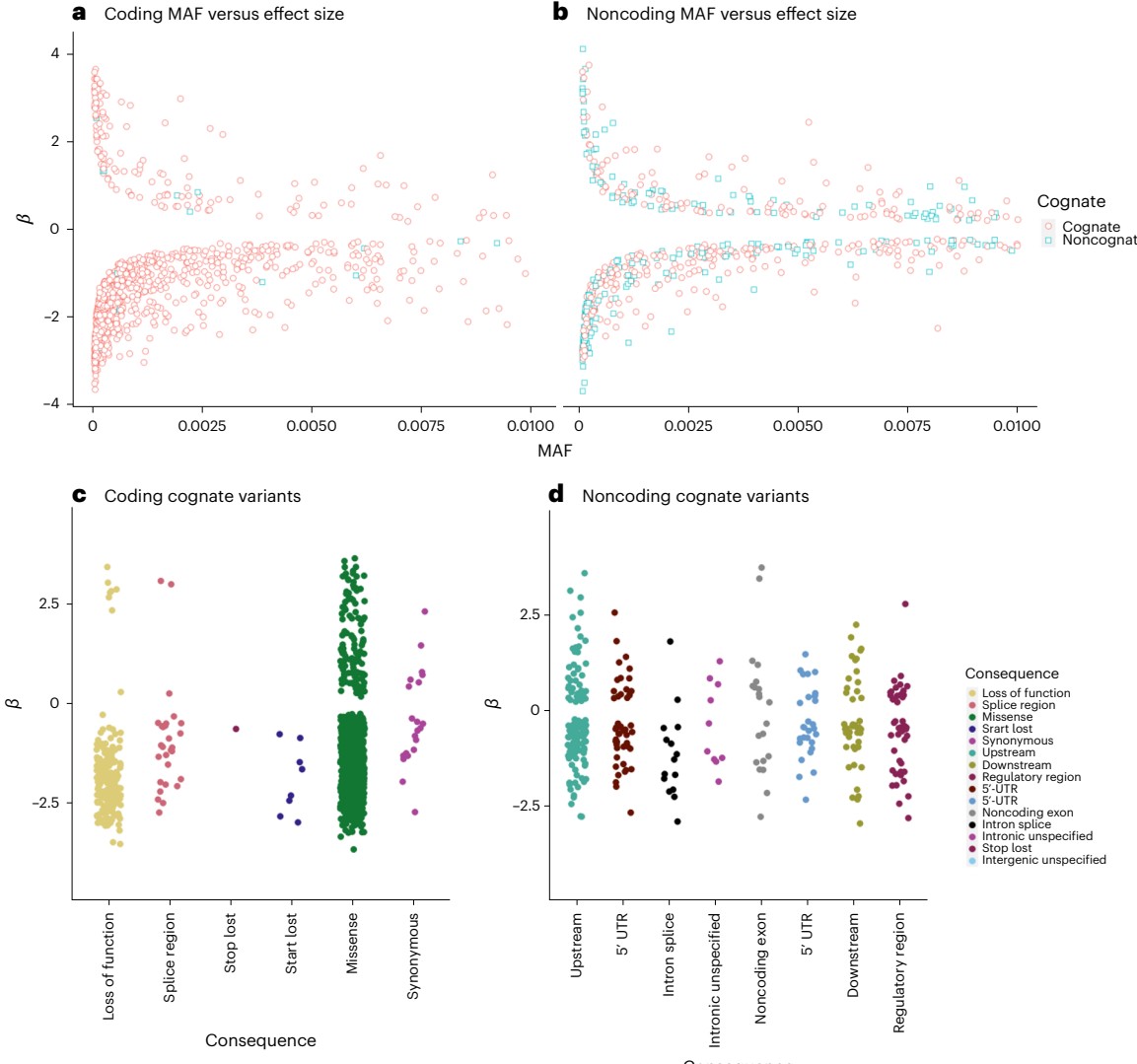

**Fig. 3 | Effect size distributions of rare, independently associated pQTL variants. a–d**, Effect sizes (β) for rare pQTL variants versus MAF, highlighting cognate (red circles) and noncognate (blue square) effects, for coding (**a**) and noncoding (**b**) pQTLs, and stratified by predicted consequence for coding variants (**c**) in the cognate gene and in the noncoding cognate gene (variants annotated as regulatory for the protein-coding gene) (**d**). The effect sizes by stratification of annotation for noncoding, noncognate variants are presented in Extended Data Fig. 3.

a relationship between the effect size of rare noncoding variants and the maximum $r^2$ with a coding pQTL ($P = 0.801$), even after adjusting for variant frequency ($P = 0.763$). This suggests little evidence of epitope effects impacting our noncoding pQTL associations.

Noncoding variants had an average absolute effect of 1.15 s.d. (median 0.86 s.d.), equating to 57.1% and 74.1% of the average absolute effect of rare loss of function and missense pQTLs, respectively. Furthermore, rare noncoding pQTLs were more evenly distributed between circulating protein-increasing and -decreasing effects (65.2% decreasing; Fig. 3d), compared with rare coding pQTLs (86.3% decreasing; $P$ heterogeneity = 5.41 × 10⁻¹³). The most strongly associated rare pQTL was within or closest to the cognate gene in 89.8% of cases. Considered separately, rare coding and noncoding pQTLs were within or closest to the cognate gene in 97.9% and 66.9% of cases, respectively. Rare noncoding pQTLs were distributed across the *cis* loci, with maximum distance from the cognate gene of 993 kb, close to the boundary of the tested *cis*-region.

We then tested for enrichment within different annotation categories for rare variants annotated as regulatory for the cognate gene. Based on the most severe predicted consequence for each variant tested, with consequences related to the cognate gene prioritized, we observed enrichment for pQTLs in the following categories: 5′-UTRs (OR = 23.6, Fisher's exact two-sided $P = 4.46 × 10^{-48}$), 3′-UTRs (OR = 3.08, $P = 4.34 × 10^{-7}$), predicted intronic splice sites (OR = 208.5, $P = 1.08 × 10^{-29}$), noncoding exons (OR = 2.69, $P = 9.02 × 10^{-5}$) and upstream variants (OR = 3.00, $P = 3.80 × 10^{-20}$; Fig. 4). We did not observe evidence of enrichment when considering rare noncoding pQTLs annotated as regulatory for another gene in the *cis*-window (Extended Data Fig. 4).

Based on the 1,651 identified rare single variants, we identified 1,040 additional unique *trans*-pQTL associations at $P < 0.05/$ (1,651 × 2,907) (Supplementary Table 11) with one of the 2,907 proteins. Note that we included *trans* effects for previously excluded proteins, because exclusion was based on *cis*-WGS data quality, not phenotype quality.

## Aggregate testing identified 357 regulatory regions
Using aggregate-based association tests, we identified 357 conditionally independent rare variant noncoding regions associated with circulating protein levels (Supplementary Tables 12 and 13). We labeled

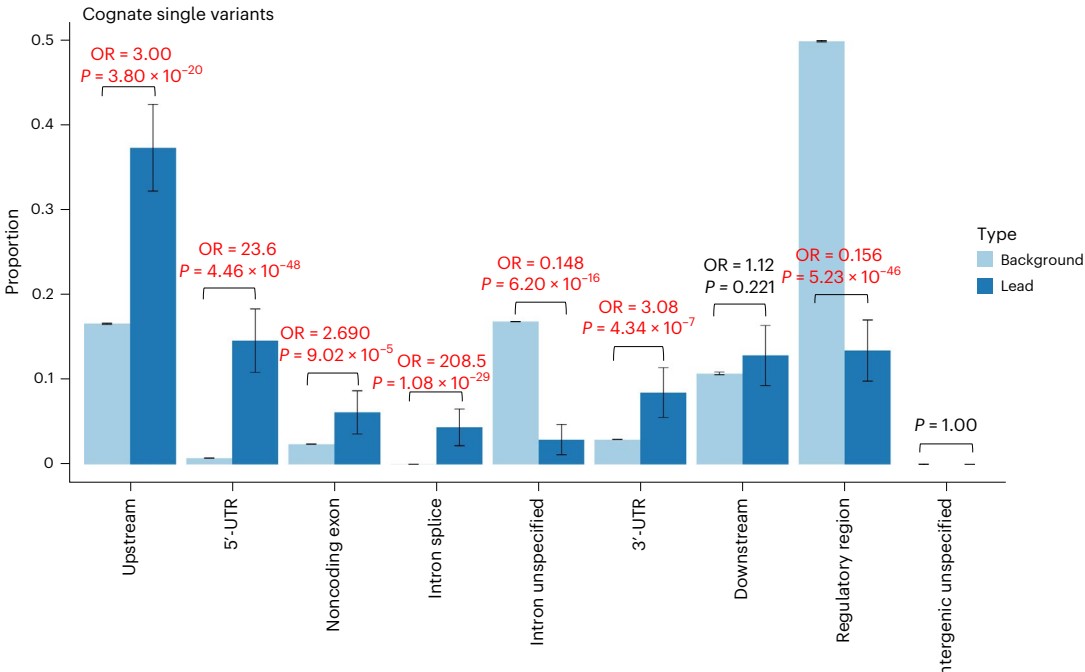

**Fig. 4 | Distribution of annotations of lead, rare, pQTL, noncoding variants compared with all variants tested.** Proportion of variants in sets of lead variants ($n = 343$; dark blue) compared with all variants tested ($N = 1,367,793$; light blue), if the variant was annotated to the cognate gene. The $P$ values are derived from a two-sided Fisher's exact test. The 95% confidence interval (CI) proportions are shown by the whiskers. The text in emboldened red indicates a comparison with evidence of enrichment or depletion at $P < 0.05/9$, where 9 is the number of statistical tests. Noncognate distributions are presented in Extended Data Fig. 4.

each independently associated aggregate association as mapping to the cognate gene if either the aggregate unit was annotated to the cognate gene itself or the start position of the aggregate lay within the cognate gene unit. All other aggregates were labeled as noncognate, although many could be closer to the cognate gene than the next nearest gene. Compared with all cognate aggregate units tested, we observed evidence of enrichment (Fig. 5) for cognate aggregates of variants annotated as 5′-UTR (OR = 6.36, $P = 5.11 \times 10^{-11}$) and predicted intronic splice (OR = 29.6, $P = 4.89 \times 10^{-17}$).

More than 90% of the noncoding aggregate signals were detected without limiting variants to those reaching our specified thresholds for conservation or constraint. Of the 357 (5.04%) conditionally independent noncoding aggregate associations, 18 were identified only when selecting highly conserved (GERP > 2) variants and 16 (4.48%) for highly constrained (JARVIS > 0.99) variants. Most of the aggregate regions would have been missed by single-variant analysis alone: 259 (72.6%) noncoding aggregates contained no lead pQTL and 74 (20.7%) contained no single variant of study-wide significance. Based on the 357 independent noncoding aggregates, we also identified 45 unique *trans*-noncoding aggregate–protein associations ($P < 0.05/(357 \times 2907)$) (Supplementary Table 14).

The vast majority of rare noncoding aggregates were identified by statistical tests allowing for bidirectional associations and for a large fraction of variants to be noncausal. Only 10 of the 357 (2.80%) rare noncoding aggregate pQTLs were most strongly associated in a burden framework (assuming that all rare variants result in effects in the same direction), in strong contrast to 38.6% of coding-based aggregates. This difference suggests that rare variants in noncoding regions are likely to result in a mixture of trait-increasing and -decreasing effects and that not all variants included in the aggregate are causal. This observation indicates that improvements to the annotation of noncoding variants will further increase the chances of detecting signals from noncoding aggregates.

We found evidence that sliding-window aggregate tests identify associations that would not have been detected with tests limited to

annotated regions. For example, there were 34 (3.31%) circulating proteins with a sliding-window pQTL. For these 34 proteins, there were 109 noncoding pQTLs when including sliding windows compared with 81 when not including sliding windows. For three proteins, the only pQTLs were sliding windows. For example, the intronic region chr3:183135000-183137000 of *LAMP3* was associated with circulating levels of LAMP3 in an ACAT-V framework ($P = 1.15 \times 10^{-13}$), but this region did not contain an Ensembl regulatory region. Our results suggest that sliding windows have added value over and above testing previously mapped regions, although it is possible that annotations present in cell- and tissue-specific data that we have not included would detect these regions.

**Rare noncoding pQTLs showed tissue-specific enrichment**

We next aimed to determine the degree to which rare noncoding pQTLs were present in tissue-relevant noncoding regulatory regions. First, we tested the hypothesis that rare noncoding pQTLs were more likely to be identified if the relevant protein were a secreted or signal protein, because circulating proteins were more likely to be representative of overall protein abundance compared with nonsecreted proteins. Second, we tested the hypothesis that rare noncoding pQTLs would be enriched in Ensembl regulatory elements for blood and liver cells ahead of 20 other tissue types (Methods) on the basis that these would be the most relevant tissue types[32]. We found that, across all regulatory regions, associations for single-variant pQTLs were highly enriched for all protein groups regardless of secretion status (Fig. 6, Extended Data Figs. 6 and 7 and Supplementary Tables 15–20). Within enhancers and CCCTC-binding factor (CTCF)-binding sites, associations were more highly enriched for secreted or signal proteins than nonsecreted proteins. Among noncoding aggregate associations, and restricting to only associations identified through sliding windows, secreted and signal proteins were more highly enriched across all regulatory regions than nonsecreted proteins. Within a sliding window, associations were most highly enriched in regulatory regions predicted active in blood vessels, followed by the liver (Supplementary Table 20).

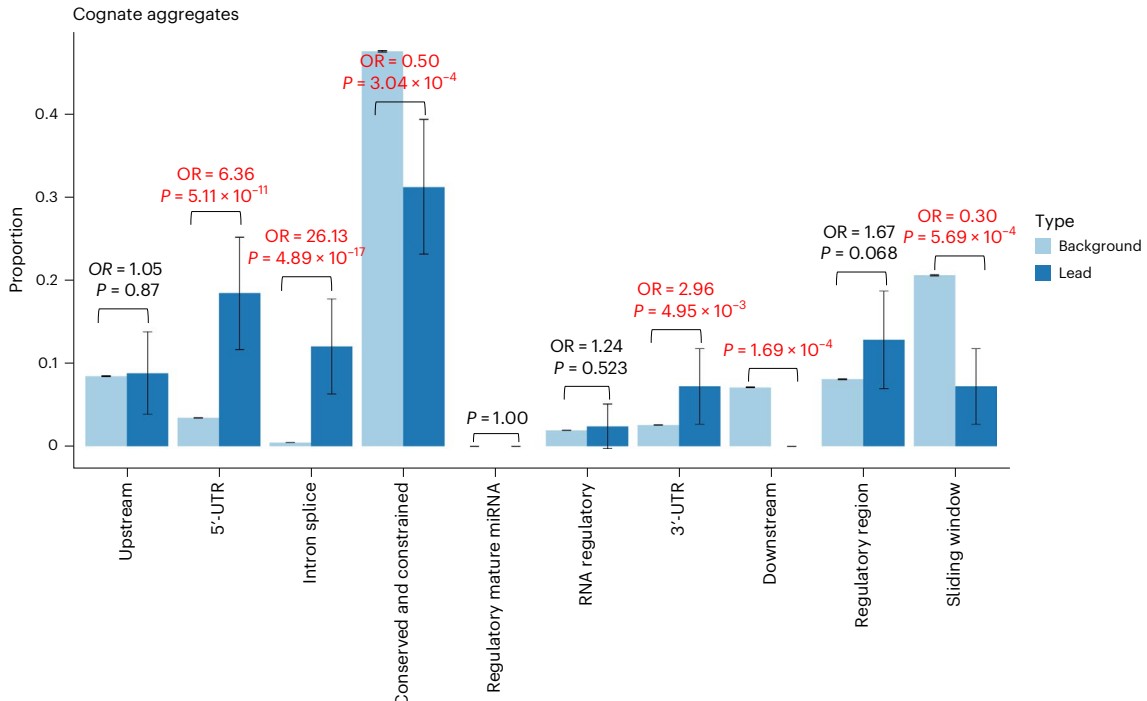

**Fig. 5 | Distribution of annotations of significant cognate aggregates of rare noncoding compared with all aggregates tested.** Proportion of aggregates in sets of lead variants ($n = 125$; dark blue) compared with all aggregates tested ($N = 1,456,986$; light blue). The $P$ values are derived from a two-sided Fisher's exact test. The 95% CI proportions are shown by the whiskers. The text in emboldened red indicates a comparison with evidence of enrichment or depletion at $P < 0.05/10$, where 10 is the number of statistical tests. Noncognate comparisons are presented in Extended Data Fig. 5.

As a sensitivity analysis, we then attempted to alleviate concerns that our discoveries may be driven by technological artefacts of the platform used to measure circulating proteins. Of the 551 proteins identified as highly concordant between the Olink (Explore 3072) and SomaScan-v.4 (ref. 15), 261 had high-quality *cis*-WGS coverage. Our pQTL associations were enriched in these 261 proteins: despite representing 25.4% of the proteins that we tested, they covered 31.1% of all our pQTLs (test of two proportions, $P = 2.85 \times 10^{-4}$), 32.7% ($P = 6.05 \times 10^{-5}$) of the rare pQTLs and 37.5% of the noncoding aggregate-based pQTLs ($P = 1.29 \times 10^{-5}$).

## Discussion

Using circulating protein levels as exemplar traits, we have shown that the analysis of WGS data enables the discovery of rare noncoding variants and aggregates associated with common phenotypes. WGS data enabled us to consider, via rare variant aggregate testing, more than double the number of variants than would have been possible through single-variant testing alone.

However, our results also showed that the number of 'statistically independent' pQTLs for a protein was inversely correlated with *cis*-sequencing coverage. Unlike previous studies, we therefore attempted to account for imperfectly captured regions: only approximately one-third of measured proteins had what we considered high-quality WGS data across the locus. Although we observed evidence for enrichment of regions previously reported as problematic within our excluded regions, demonstrating that these issues are not unique to UKB, further analysis is required to fully understand the observed low coverage in each of the *cis* regions. This is a crucial point of warning for others who will be using sequencing data at this scale.

We have identified hundreds of new noncoding rare aggregate and single-variant associations with measured protein levels in *cis*-windows 1 Mb either side of the cognate gene. We show that the effect sizes of noncoding associations sometimes have similar magnitude comparable to coding associations but more balanced between

protein-increasing and -decreasing effects. As a result of the complex nature of LD and haplotype effects, it is extremely difficult to determine whether a coding signal is driving a noncoding signal or vice versa. To mitigate against this explanation, we took a conservative approach and conditioned on all coding variants for the cognate gene.

We observed some differences in number of single-variant pQTLs per protein between our results and those of Eldjarn et al.[15]. These differences may be partly driven by methods for conditional analysis: their analysis used forward stepwise conditional analysis to define conditionally independent pQTLs, the issues related to which have been previously detailed (for example, ref. 33), whereas we performed both forward- and backward-conditional analysis steps implemented in GCTA-CoJo[19]. These differences in pQTLs between the two studies highlight the difficulties of interpreting multiple independent pQTLs at the same locus.

We demonstrate that the 5′-UTR and predicted intronic splice acceptor or donor sites were enriched for rare noncoding pQTLs. As UTRs and introns are not typically captured in exome sequencing, which targets coding exons, our results highlight the importance of WGS for finding new rare gene-centric variants.

We additionally demonstrate the power of aggregate testing for noncoding regions, analogous to well-established methods aggregating functionally similar variants in coding regions. By testing rare genomic aggregates of noncoding elements, grouped by, for example, proximity to genes or predicted regulatory activity, or using sliding windows, we identified a further 464 conditionally independent regions not identified by single-variant testing alone.

Compared with aggregate-based coding associations, noncoding genomic aggregate pQTLs were enriched for tests allowing bidirectional effects and/or sparse causality. This observation is consistent with the fact that prediction of variant effects and functional regions is less precise in noncoding regions. However, the fact that we have identified noncoding associations with current annotations and data indicates that more discoveries in common phenotypes are

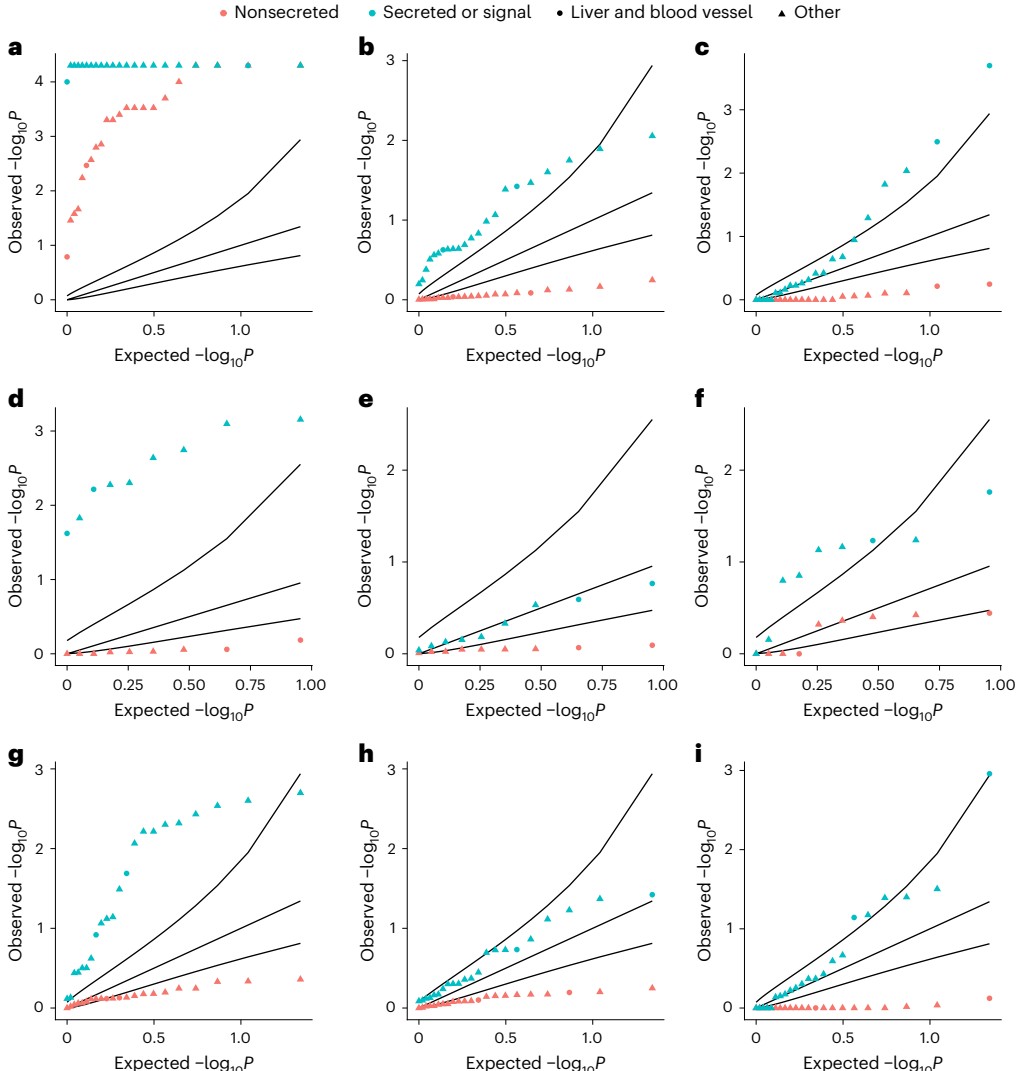

**Fig. 6 | QQ plot for enrichment of loci within Ensembl-predicted active regions within tissue groups.** Empirical one-sided *P* values for enrichment of signals within Ensembl-predicted active regions in 22 tissue groups for proteins by secretion status. **a**,**d**,**g**, Enrichment for single variants. **b**,**e**,**h**, Enrichment for Ensembl regulatory region-based aggregate tests. **c**,**f**,**i**, Enrichment for sliding-window-based aggregate tests. **a**–**c**, Enrichment within all predicted active regions. **d**–**f**, Enrichment for promoters. **g**–**i**, Enrichment for enhancers. The black curves represent, from top to bottom per plot, the 95% CIs of the expected *P*-value distribution under the null hypothesis. The proteins in each annotation category can be found in Supplementary Table 21. *P* values were derived from the proportion of sampled variants with equal or higher overlap with annotated regions. The blue points indicate secreted or signal proteins and the red ones nonsecreted protein groups. Liver and blood vessel cells are indicated by circles and all other cell types by triangles.

likely as functional annotations improve and population genetic data accumulate.

There were some limitations to our study. First, we were unable to replicate our results in an independent dataset because we did not have access to similar data from other studies. However, a large proportion of our associations reached levels of statistical confidence far beyond our threshold. Furthermore, although our primary analysis was limited to individuals of European ancestry because of sample size restrictions, effect sizes were consistent across ancestry groups. Second, we cannot be certain that we have accounted for all possible sources of residual confounding by LD with coding or common variants, including variants in complex or repetitive regions. For similar reasons, the pQTLs that we have identified in the same region may not be truly independent. However, it is unlikely that our associations are substantially affected by residual confounding from coding variants because they have different features, including the more equal distribution between protein-increasing and -decreasing effects. We also observed no evidence of a relationship between the effect sizes of independent noncoding pQTLs and residual LD with coding variants, suggesting that our results are unlikely to be confounded by epitope effects. Third, all circulating proteins were measured in blood. Although a large portion of tissue-specific proteins are expressed only in those specific tissues, we were limited to considering circulating protein levels. Fourth, we were unable to take account of binding effects related to the technology used to measure protein levels. However, we did observe an enrichment of associations within the subset of 551 gold standard proteins shown to correlate strongly between Olink and SomaLogic platforms[15]. Finally, although we state that sliding-window aggregates identified potentially new regulatory regions, it was not possible to exhaustively examine all publicly available regulatory maps. Furthermore, testing all possible tissue-specific annotations incurs a large hypothesis-testing burden.

In conclusion, using an exemplar trait of circulating protein measurements, we have found multiple new associations between circulating protein levels and rare noncoding variants. Our results indicate that there are likely to be many rare noncoding variants with large effects on complex phenotypes waiting to be discovered.

## Online content

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

## Methods

This research complies with all appropriate ethical regulations. Ethics approval for the UKB study was obtained from the North West Centre for Research Ethics Committee (protocol no. 11/NW/0382).

### UKB and WGS

The WGS performed for UKB had an average coverage of 32.5× using Illumina NovaSeq 6000 sequencing machines[34] with 150-bp paired-end reads. The genome build used for sequencing was GRCh38: single-variant nucleotide polymorphisms and short indels were jointly called using DRAGEN 3.7.8 (ref. [34]).

### Human protein expression levels

Protein levels of 2,932 proteins for up to 54,219 UKB participants were profiled using Olink technology, as described in ref. [17], by the UKB Pharma Proteomics Project. Quality control procedures were applied to the data before being made available for researcher use, including outlier removal. Protein levels were additionally $\log_2$(transformed) before release. After quality control filtering, 54,189 individuals with protein expression data were approved for analysis. Sun et al.[17] found no evidence of batch- or plate-confounding effects.

### Genetic data filtering

We set any DRAGEN WGS genotype calls to missing if either the sum(LAD) < 8 (where LAD is local allele depth) per sample genotype or genotype quality (GQ) < 10 for each of the 154,430 project variant call formats (pVCFs) provided by UKB using bcftools v.1.2 (ref. [35]). After these additional quality control steps, the transmission rate of singletons, which should theoretically be exactly 0.5 (assuming that most variants are not under strong negative selection), was 0.497, compared with 0.456 as originally provided by UKB. We subsequently dropped any variant with a missingness >10%. A multi-allele splitting procedure was applied and each variant was assigned a unique ID (CHR:BP:REF:ALT) before merging all VCFs per chromosome and indels were normalized and left aligned using bcftools based on a 1000G b38 reference available at https://ftp.1000genomes.ebi.ac.uk/vol1/ftp/technical/reference/GRCh38_reference_genome (accessed 30 March 2024). Each merged pVCF was then converted to plink[36] (v.2.0) p(gen/var/sam) format.

### Structural variants

Structural variants (variants ≥50 bp) for the DRAGEN release were available only in single-sample format at the time of analysis. We used SURVIVOR v.1.07 (ref. [37]) with default settings to merge and harmonize structural variant calls into a pVCF for the 54,219 individuals with circulating protein measures. We then converted that project (p)VCF into plink p(gen/var/sam) format and merged with the DRAGEN SNV or indel call plink files.

### Genetic variant annotation

We annotated all genetic variants using Ensembl VEP (v.110), LOFTEE[30] and UTRannotator[38]. Where possible, we assigned each variant to one of three classifications: coding, proximal regulatory or intergenic regulatory. A variant was classified as coding if it had a predicted impact on the coding sequence of any transcript, proximal regulatory if the variant lay within a 5-kbp window of the UTRs of a transcript and was not already a coding variant in any transcript and finally intergenic or intronic regulatory if it was not coding and was mapped to a gene-agnostic regulatory element (details below). We additionally tested variants in sliding windows of size 2,000 bp, regardless of the number of variants in each window, with coding variants excluded to minimize hypothesis overlap.

We then assigned each variant to groupings, which we refer to as masks, according to their predicted consequence and location. We used five published variant scores to group variants by consequence: (1) GERP: the GERP score is a measure of conservation at the variant level[22]. We classified a variant as highly conserved if it had a GERP score >2. (2) PhastCon score: phastCon is a window-based measure of conservation across species[39], either strictly mammalian (phastCon 30) or for all species (phast_100). We tested noncoding genome windows, that is, excluding any window containing an exon, that had a phastCon score in the 99th percentile. (3) Constrained score: constraint was calculated in windows of size 1 kbp (ref. [6]) based on the local mutability and observed mutation rate of each window. We tested windows with a constraint $z$-score ≥4. (4) SpliceAI score: the SpliceAI score[40] is a measure of how well predicted each variant within a pre-messenger RNA region is of being a splice donor or acceptor, or neither. A variant was classified as a splice site with high confidence if it had an AI > 50. (5) CADD: the CADD score[23] predicts how deleterious a variant is likely to be. We applied the CADD score only to coding variants and considered loss-of-function variants only if tagged as high confidence by VEP. Missense variants with CADD > 25 were segregated for testing in a separate mask. (6) JARVIS score: the JARVIS score was derived to better prioritize noncoding genetic variation for association study, based on a machine learning model derived from measures of constraint[21].

Each genome mask consisted of a number of variants with different consequences, based on their location, one of the above scores and/or predicted coding consequences. For example, for a variant to be classified as missense CADD > 25, it must change a codon of an exon of a gene transcript and be predicted to be highly deleterious. In Supplementary Table 22, we present the full list of consequences assigned to each mask and classification.

### Association analyses

We performed both single-variant and aggregate tests within *cis*-loci for each of the 2,932 proteins measured in UKB. To define the *cis*-window, we first mapped each protein to a coding gene (see Supplementary Table 1 for a small number of exclusions) and for each gene determined the longest transcript recorded by Ensembl. Based on the longest transcript, we then defined the *cis*-window as a 1-Mb window either side of the 5′- and 3′-UTRs of the transcript gene (limited by the beginning and ends of chromosomes), as well as the variants within the coding and intronic sequences. All association analyses were corrected for age, sex and age$^2$, UKB recruitment center (as a proxy for geography) the first 40 genetic PCs, WGS batch, Olink plate, fasting time and time since blood draw.

### Single-variant association testing

To identify *cis* single variants associated with protein levels, we first performed an association test for all genetic variants with an MAC of at least five using regenie v.3.3 (ref. [41]) in the *cis*-window. Lead variants were then selected in a conditional-joint analysis using an altered version of GCTA-CoJo[19] (diff-freq = 0.2, cojo $P = 2.95 \times 10^{-10}$), with the UKB WGS data, limited to individuals with proteomic data, as an LD reference panel. Testing revealed that GCTA-CoJo filters variants if their variance explained is >900× the smallest variance explained by any independent variant ('sqrt(ldlt_B.vectorD().maxCoeff()/ldlt_B.vectorD().minCoeff()) > 30': line 732 of gcta/meta/joint_meta.cpp at https://github.com/jianyangqt/gcta (accessed 17 March 2024)). We understand that this filter exists to capture statistical confounding caused by collinearity, for example, if the reference genome used to calculate LD and the genetic data do not correlate well. However, for our purposes of jointly considering common and rare variants, where we have used the exact LD reference panel matching our discovery dataset, we found that this filter was falsely removing large-effect pQTLs. We thus removed this filter and re-compiled GCTA-CoJo, which is available at https://github.com/ExeterGenetics/WGS_50k_Proteins_2024.

GCTA-CoJo assumes $P = 0$ beyond a certain threshold. Where this has occurred, we have re-calculated $\log_{10}(P)$ according to the following R-script: $\log_{10}(P\text{-calculated}) = \log_{10}(\exp(1)) \times (pt(-abs(beta/s.e.), d.f. = N-2, \log = TRUE) + \log(2))$.

## UKB WGS coverage calculation and filtering

We calculated coverage from the UKB WGS data, based on our quality control criteria, for 998 inferred European ancestry individuals with both proteomic data and WGS. We first calculated the number of samples at each base-pair with depth >8. We subsequently calculated the percentage of bases between the 5′-UTRs and 3′-UTRs for each cognate protein-coding gene that had >8 depth for >90% of individuals. We repeated this analysis for the full *cis*-window. A protein's gene-level coverage and region-level coverage were then both required to have >99.5% of bases and >90% of individuals with depth > 8.

To account for single damaging coding bases being missed or poorly genotyped, we additionally filtered proteins if the maximum difference between the marginal (from our regenie analysis) and joint (GCTA-CoJo) effect estimate ($\beta$), for any pQTL associated with that protein, was beyond the 90th percentile. A flow chart describing the complete filtering procedure is provided in Extended Data Fig. 8.

We then tested for overlap between our results and genomic regions excluded for low coverage with a database of problematic genomic regions, generated by Genome in a Bottle and the precision-FDA Truth Challenge v.2 (refs. 28,29) (release v.3.5, accessed 10 September 2024). These included "tandem repeats, all homopolymers >6 bp, all imperfect homopolymers >10 bp, all difficult to map regions, all segmental duplications, GC < 25% or >65%, 'Bad Promoters', chrX/Y XTR and ampliconic, satellites and 'OtherDifficult' regions (including regions from the T2T-consortium for GRCh38 only)"[28,29]. For each protein, we determined the number of problematic regions of each type which shared at least 1-bp overlap with either the gene region (5′- to 3′-UTR) or 2-Mbp window. These variables were tested in a logistic model against a binary variable describing whether the protein had been filtered for low coverage, adjusting for the total length of the 2-Mbp window (which naturally varies by the gene size).

## Rare variant genomic aggregate testing

To identify noncoding, potentially regulatory regions of the genome that were insufficiently powered for single-variant analysis, we subsequently performed noncoding rare variant (MAF < 0.1%) genomic aggregate association tests. To test whether noncoding rare variant aggregate signals were caused or confounded by residual LD and haplotype structure with common variants and/or single-variant signals, we performed the following steps for each rare variant aggregate test result reaching Bonferroni's $P < 0.05$: (1) to generate our primary noncoding discovery results we adjusted for the common lead variants identified as independent signals in the joint (CoJo) analysis (at MAF > 0.1%, ~MAC = 40) and adjusted for all genetic variants (regardless of $P$ value) which we had annotated as coding in any transcript of the gene that mapped to the protein of interest; (2) to identify independent noncoding aggregate associations, if at least one aggregate passed our significance threshold we performed a forward stepwise regression. Starting from the most strongly associated noncoding aggregate (by $P$ value), per protein, we performed an additional noncoding aggregate testing run on aggregates reaching genome-wide significance, adjusting for all variants in the top signal. This process is repeated, with more variants added from the next most strongly associated aggregate, until no aggregate is genome-wide significant. (3) To establish the extent to which our primary aggregate discovery results could be the result of a single low-frequency lead variant, we identified aggregate associations containing exactly one lead genetic variant. (4) As a sensitivity step, to establish the extent to which these results could be caused by confounding LD, we performed a further step where we adjusted for all pQTL single variants identified

Gene unit testing was performed for variants with a maximum allele frequency threshold of 0.1%, using regenie, based on the genetic units specified in Supplementary Table 22. Regenie performs four types of genome unit tests: (1) standard BURDEN tests, under the assumption that each variant in a given gene unit mask has approximately the same effect size and sign on the phenotype; (2) SKAT tests, where the sign of association of each variant in the unit is allowed to vary; (3) ACAT tests, where the sign of association of each variant in the unit can differ and only a small number of variants in the mask need be associated; and (4) ACAT-O, which is an omnibus test of BURDEN, SKAT and ACAT that aims to maximize the statistical power across the three tests.

We performed each of the four statistical tests above for each mask for which a gene unit has at least one variant. In addition, an association test was performed for all singleton variants (with MAC = 1) in each unit; regenie also estimated an 'all-mask' association strength for each genome unit, which is an aggregation of the test statistics of the individual masks. To ensure that this did not result in a mixing of non-coding and coding association statistics, we split each gene transcript into a coding transcript, which we tested for all coding masks, and a proximal transcript, which we tested for all proximal masks. Regulatory genome units were classified by their ENSR assignment, the extent of a 1-kb constrained window or a phastCon-conserved window. We named sliding-window masks by the region of the respective chromosome that they covered. The code required to perform the primary analysis is available at Zenodo[42].

## Statistical significance

Statistical significance was defined based on the minimum $P$ value observed for a WGS analysis of 20 randomly generated normally distributed continuous traits. The minimum $P$ values for single-variant and aggregate association analyses were treated as independent: $P$ (single variants) = $2.95 \times 10^{-10}$; $P$ (aggregates) = $8.71 \times 10^{-9}$.

## Ensembl regulatory region enrichment

We calculated an enrichment in overlap for both single variants and aggregate regions with Ensemble[31] regulatory regions, which are available for 118 tissues or cell lines, compared with the genetic background. For each tissue, Ensembl additionally provides predictions on whether each region is active (or inactive, suppressed and so on), and the type of regulatory activity (promoter, enhancer, CTCF-binding site, transcription-factor-binding site or open chromatin region). We subsequently exclusively considered regions that were predicted to be active, excluded cell lines and cancer-derived tissues and grouped the remaining tissue types into 22 supergroups (Supplementary Table 15). We grouped proteins into those annotated as secreted, signal or membrane bound using the DAVID software[43,44] (Supplementary Table 21). We then grouped proteins as secreted or signal proteins if they were annotated into either group and labeled all other proteins as nonsecreted.

To determine the statistical enrichment, we performed bootstrapping over 10,000 simulations. For each simulation, we randomly selected a number of rare noncoding variants or aggregates equal to the number of independent signals determined by the set of our rare noncoding single-variant or aggregate tests from the *cis*-regions of the genome that we tested for association. We then determined the overlap of the randomly selected set of variants or aggregates with any of the regulatory regions (we reperformed this for each stratum of panel and tissue type) and compared the distribution of the number of overlaps for any simulation with the number overlapping in our independent associations. We then assigned an empirical $P$ value to the observed overlap.

## Statistics and reproducibility

The present study included all individuals in UKB who also had measured protein levels and whole-genome sequences. Participants were aged between 37 and 80 years at study recruitment. PC analysis of genetic data was used to define homogeneous groups based on genetic ancestry. Participants were genetically grouped into European-like, south Asian-like and African-like genetic ancestry. Analysis was performed separately within each group adjusting for PCs in a mixed model

to account for cryptic relatedness. The present study was observational, so neither randomization nor binding was applicable.

## Reporting summary

Further information on research design is available in the Nature Portfolio Reporting Summary linked to this article.

## Data availability

Data cannot be shared publicly because of the data availability and data return policies of the UKB. Data are available from the UKB for researchers who meet the criteria for access to their datasets (http://www.ukbiobank.ac.uk). Summary statistics are available via Zenodo at https://doi.org/10.5281/zenodo.14203628 (ref. 42).

## Code availability

Analysis code relating to the analyses is available via GitHub (https://github.com/ExeterGenetics/WGS_50k_Proteins_2024).

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

## Acknowledgements

This manuscript is part of the Stratification of Obesity Phenotypes to Optimize Future Obesity Therapy (SOPHIA) project. SOPHIA has received funding from the Innovative Medicines Initiative 2 Joint Undertaking (grant no. 875534). This Joint Undertaking received support from the European Union's Horizon 2020 research and innovation program and EFPIA and T1D Exchange, JDRF and Obesity Action Coalition (www.imisophia.eu). G.H. received funding from the Innovative Medicines Initiative 2 Joint Undertaking (grant no. 875534). A.R.W. was supported by the Academy of Medical Sciences, the Wellcome Trust, the Government Department of Business, Energy and Industrial Strategy, the British Heart Foundation and the Diabetes UK Springboard Award (no. SBF006\1134). The research utilized data from the UKB resource carried out under UKB application no. 103356. UKB protocols were approved by the National Research Ethics Service Committee. T.M.F. was supported by the Medical Research Council (MRC) awards (nos. MR/WO14548/1 and MR/T002239/1). M.N.W. and R.N.B. were supported by the MRC (grant no. MR/Y003748/1). We acknowledge the use of the University of Exeter High-Performance Computing facility in carrying out this work, funded by an MRC Clinical Research Infrastructure award (grant no. MR/M008924/1). The present study was supported by the National Institute for Health and Care Research (NIHR) Exeter Biomedical Research Centre. The views expressed are those of the authors and not necessarily those of the NIHR or the Department of Health and Social Care. This communication reflects the author's view: IMI, the European Union, EFPIA or any associated partners are not responsible for any use that may be made of the information contained therein.

## Author contributions

G.H., R.N.B., M.N.W. and T.M.F. conceived the study. G.H, R.N.B. and K.C. performed the analyses. G.H., R.N.B., K.C., L.J., A.M., K.A.P., T.M.F., C.F.W., A.R.W. and M.N.W. wrote and edited the paper. All authors have read and approved the paper.

## Competing interests

The authors declare no competing interests.

## Additional information

**Extended data** is available for this paper at https://doi.org/10.1038/s41588-025-02095-4.

**Correspondence and requests for materials** should be addressed to Gareth Hawkes, Michael N. Weedon, Timothy M. Frayling or Robin N. Beaumont.

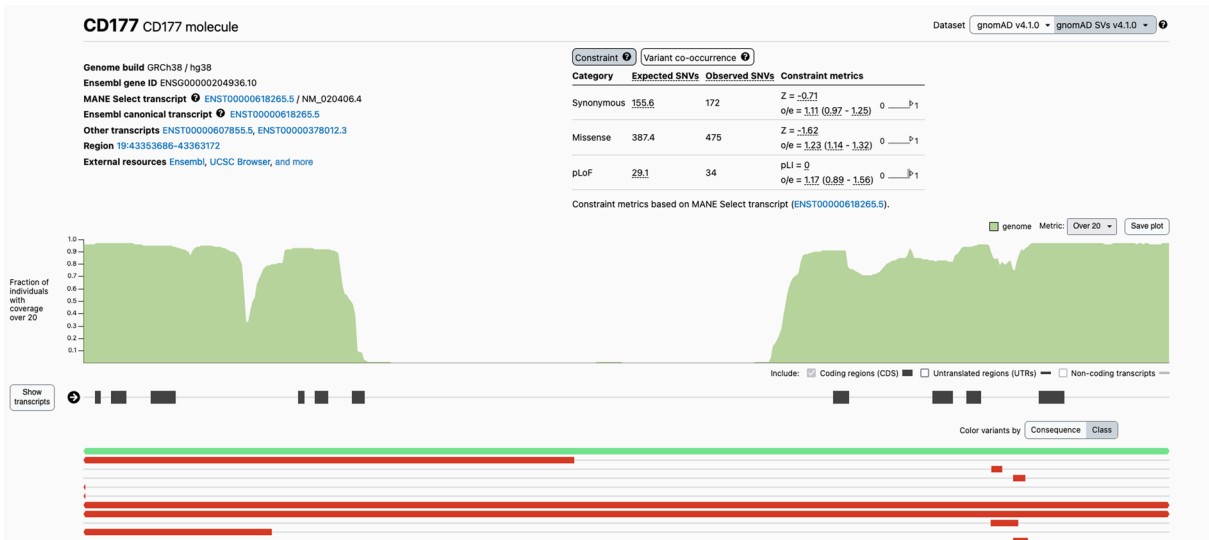

**Extended Data Fig. 1 | Depletion of whole genome sequencing coverage over CD177.** GNOMAD browser v4.1 view showing depleted coverage over exon 5 and intron 6 of CD177.

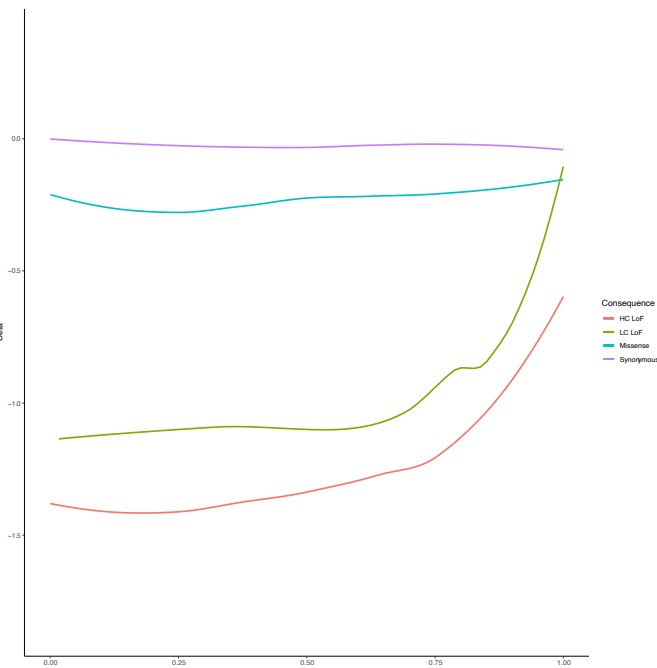

**Extended Data Fig. 2 | Distribution of effect scores by relative gene positions for four primary coding consequence annotations.** Stratified into high-confidence loss-of-function (HC LoF), low-confidence loss-of-function (LC LoF), missense and synonymous predicted consequences. Error bands (grey) represent 95% confidence intervals.

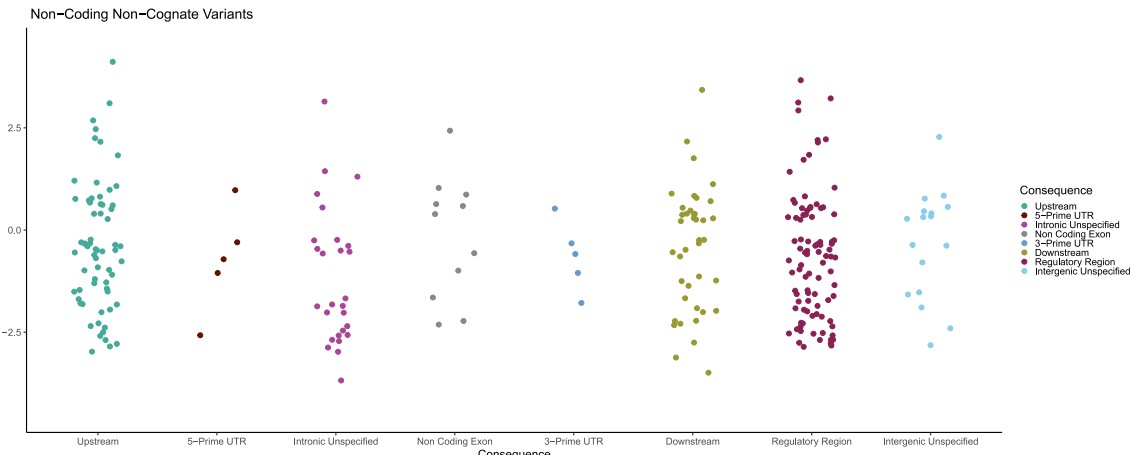

**Extended Data Fig. 3 | Effect size distributions of rare independently associated non-cognate pQTL variants.** Effect sizes for rare non-cognate pQTL variants stratified by predicted consequence.

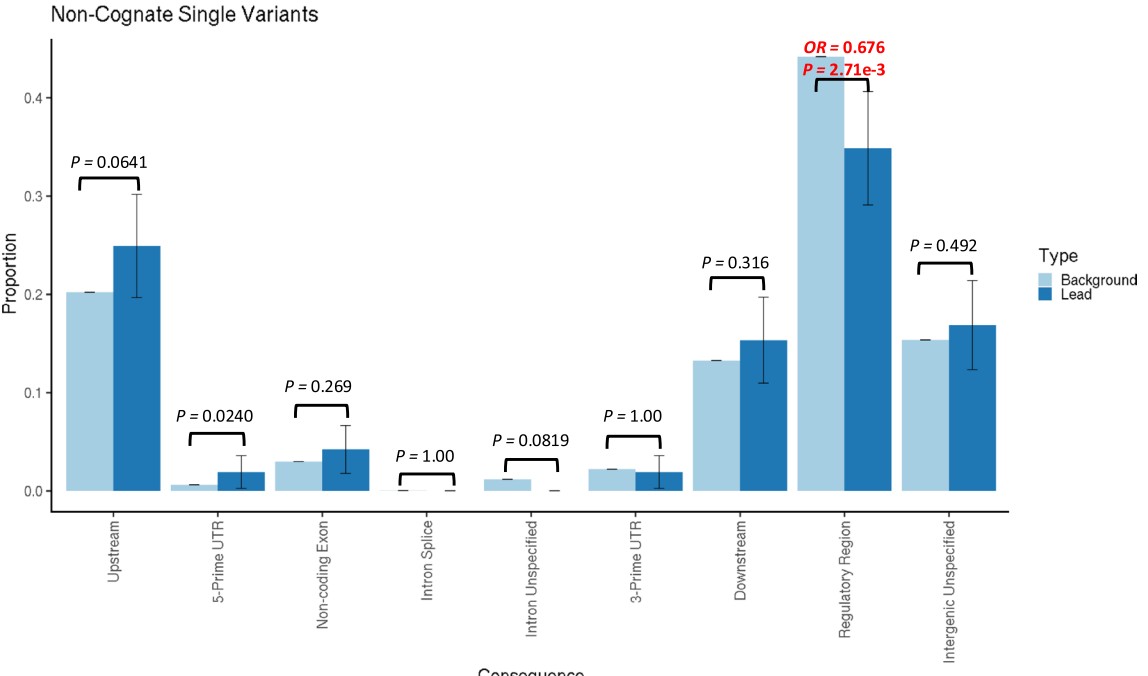

**Extended Data Fig. 4 | Distribution of annotations of lead rare pQTL non-coding non-cognate variants compared to all variants tested.** Proportion of variants in sets of lead variants (N = 261; dark blue) compared to all variants tested (N = 35,120,659 light blue), if the variant was not annotated to the cognate gene. 95% confidence intervals of proportions shown by whiskers. p-values are derived from a two-sided Fisher's exact test. Text in bolded red indicates a comparison with evidence of enrichment or depletion at P < 0.05/9.

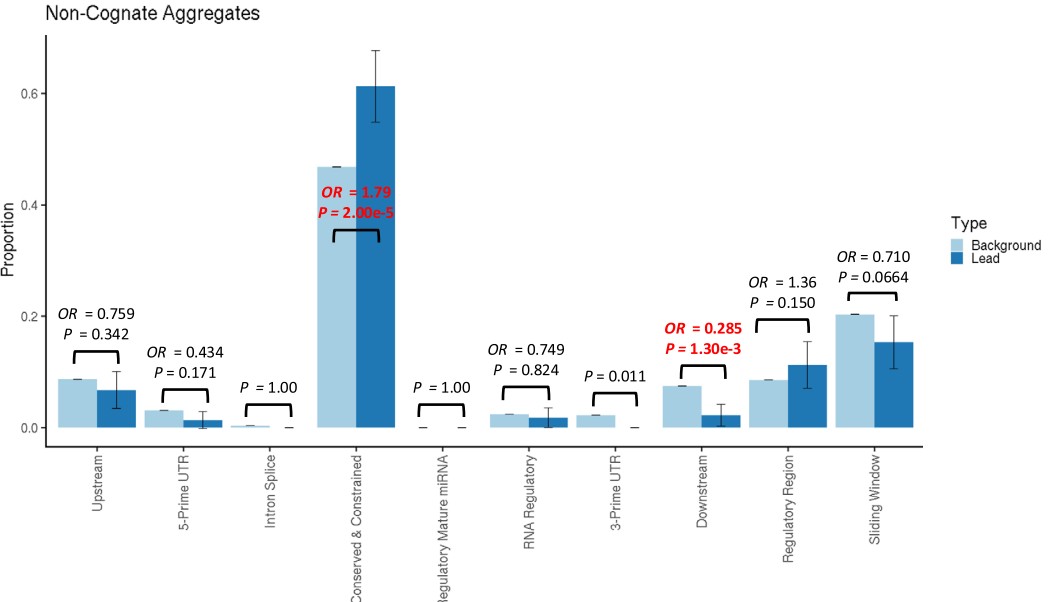

**Extended Data Fig. 5 | Distribution of annotations of significant non-cognate aggregates of rare non-coding compared to all aggregates tested.** Proportion of aggregates in sets of lead variants (N = 222; dark blue) compared to all aggregates tested (N = 34,657; 962 light blue). 95% confidence intervals of proportions shown by whiskers. p-values are derived from a two-sided Fisher's exact test. Text in bolded red indicates a comparison with evidence of enrichment or depletion at $P < 0.05/10$.

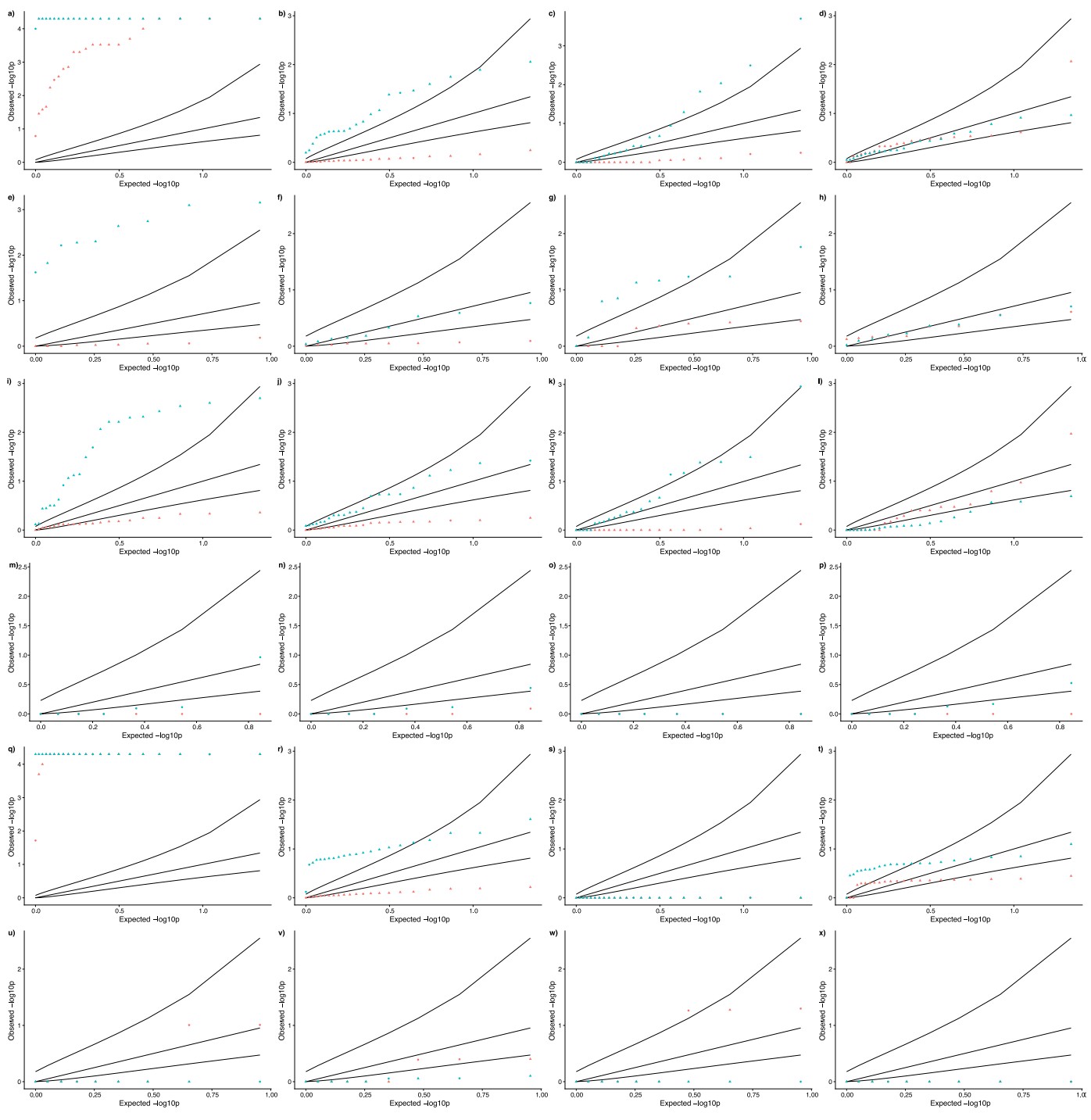

**Extended Data Fig. 6 | QQ plot for enrichment of loci within Ensembl predicted active regions within tissue groups.** Empirical one-sided P-values for enrichment of signals within Ensembl predicted active regions within tissue groups. Panels **a**), **e**), **i**), **m**), **q**), and **u**) show enrichment for single variants, panels **b**), **f**), **j**), **n**), **r**), and **v**) show enrichment for aggregate tests, panels **c**), **g**), **k**), **o**), **s**), and **w**) show enrichment for sliding-window based aggregate tests, and panels **d**), **h**), **l**), **p**), **t**), and **x**) show enrichment for Ensembl regulatory region based aggregate tests. Panels **a**), to **d**) show enrichment within all predicted active regions, panels **e**) to **h**) show enrichment within CTCF binding sites, panels **i**) to **l**) for enhancers, **m**) to **p**) for open chromatin regions, **q**) to **t**) for promoters, and **u**) to **x**) for transcription factor binding sites. P-values were derived from the proportion of sampled variants with equal or higher overlap with annotated regions.

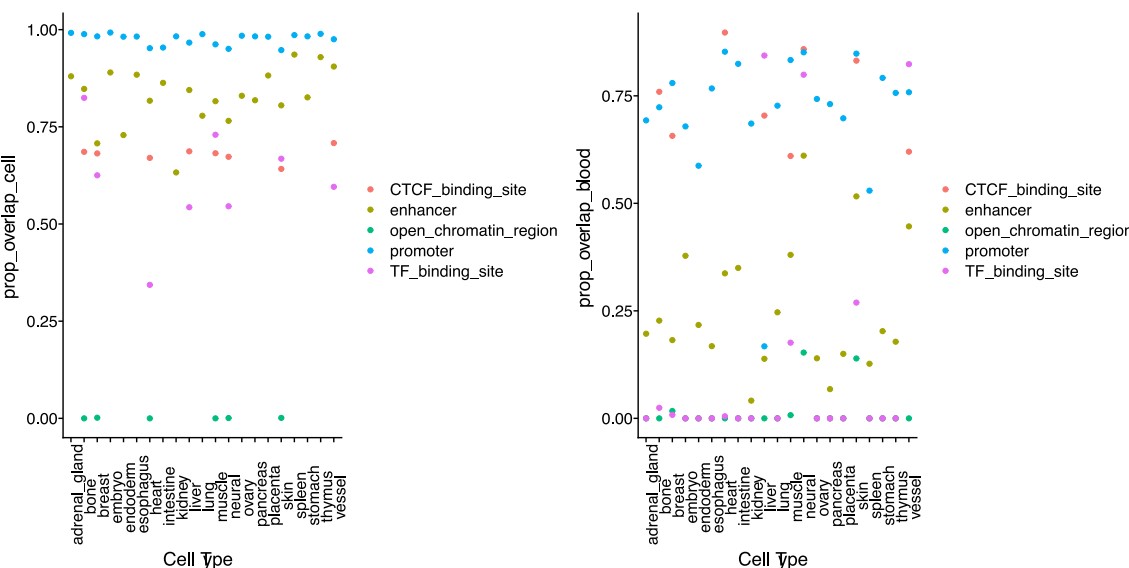

**Extended Data Fig. 7 | Proportion of Ensembl predicted active regions shared between blood and other tissues.** Left panel: proportion of active regions overlapping between blood and other tissues as a proportion of the active regions for that tissue. Right panel: proportion of active regions overlapping between blood and other tissues as a proportion of the active regions in blood.

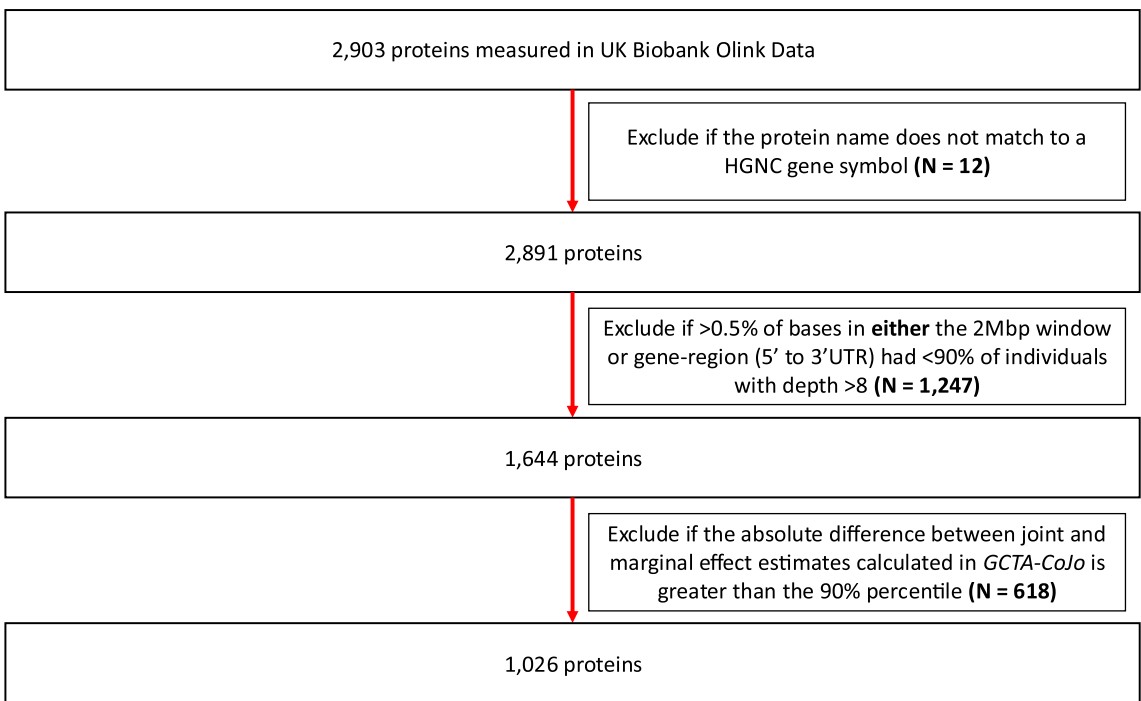

**Extended Data Fig. 8 | Flow diagram for protein exclusion.** Flow diagram describing the procedure by which proteins were excluded from our analyses.

Robin N Beaumont
Timothy M Frayling
Michael N Weedon

# Reporting Summary

## Statistics

For all statistical analyses, confirm that the following items are present in the figure legend, table legend, main text, or Methods section.

| n/a | Confirmed | |
|---|---|---|
| ☐ | ☒ | The exact sample size ($n$) for each experimental group/condition, given as a discrete number and unit of measurement |
| ☒ | ☐ | A statement on whether measurements were taken from distinct samples or whether the same sample was measured repeatedly |
| ☐ | ☒ | The statistical test(s) used AND whether they are one- or two-sided<br>*Only common tests should be described solely by name; describe more complex techniques in the Methods section.* |
| ☐ | ☒ | A description of all covariates tested |
| ☐ | ☒ | A description of any assumptions or corrections, such as tests of normality and adjustment for multiple comparisons |
| ☐ | ☒ | A full description of the statistical parameters including central tendency (e.g. means) or other basic estimates (e.g. regression coefficient) AND variation (e.g. standard deviation) or associated estimates of uncertainty (e.g. confidence intervals) |
| ☐ | ☒ | For null hypothesis testing, the test statistic (e.g. $F$, $t$, $r$) with confidence intervals, effect sizes, degrees of freedom and $P$ value noted<br>*Give P values as exact values whenever suitable.* |
| ☒ | ☐ | For Bayesian analysis, information on the choice of priors and Markov chain Monte Carlo settings |
| ☒ | ☐ | For hierarchical and complex designs, identification of the appropriate level for tests and full reporting of outcomes |
| ☒ | ☐ | Estimates of effect sizes (e.g. Cohen's $d$, Pearson's $r$), indicating how they were calculated |

*Our web collection on statistics for biologists contains articles on many of the points above.*

## Software and code

Policy information about availability of computer code

| Data collection | NA |
|---|---|
| Data analysis | Variant annotation was performed using VEP version 110. Association analysis was performed using Regenie v3.3. R-analysis was performed using v4.1.2. Plink files were generated using v1.9 and v2.0 as appropriate. All other analysis code was custom written and will be released in a github repository |

For manuscripts utilizing custom algorithms or software that are central to the research but not yet described in published literature, software must be made available to editors and reviewers. We strongly encourage code deposition in a community repository (e.g. GitHub). See the Nature Portfolio guidelines for submitting code & software for further information.

## Data

Policy information about availability of data

All manuscripts must include a data availability statement. This statement should provide the following information, where applicable:
- Accession codes, unique identifiers, or web links for publicly available datasets
- A description of any restrictions on data availability
- For clinical datasets or third party data, please ensure that the statement adheres to our policy

Participant-level data cannot be shared publicly because of data availability and data return policies of the UK Biobank. Data are available from the UK Biobank for

researchers who meet the criteria for access to datasets to UK Biobank (http://www.ukbiobank.ac.uk). Summary statistics are provided in the supplementary tables (GRCh38).

## Research involving human participants, their data, or biological material

Policy information about studies with human participants or human data. See also policy information about sex, gender (identity/presentation), and sexual orientation and race, ethnicity and racism.

| | |
|---|---|
| Reporting on sex and gender | Findings apply to both sexes. Sex was used as a covariate in the analyses and males and females were analysed together |
| Reporting on race, ethnicity, or other socially relevant groupings | Principal Component Analysis of genetic data was used to define homogeneous groups based on genetic ancestry. Analysis was performed separately within each group adjusting for PCs in a mixed model to account for cryptic relatedness |
| Population characteristics | Participants were genetically grouped into European-like, South Asian-like and African-like genetic ancestry. Participants were aged between 37 and 80 at study recruitment. |
| Recruitment | Participants were recruited to the UK Biobank study through self selection as described in central UK Biobank resources (http://www.ukbiobank.ac.uk) |
| Ethics oversight | UK Biobank protocols were approved by the National Research Ethics Service Committee. TMF is supported by MRC awards MR/WO14548/1 and MR/T002239/1. |

Note that full information on the approval of the study protocol must also be provided in the manuscript.

# Field-specific reporting

Please select the one below that is the best fit for your research. If you are not sure, read the appropriate sections before making your selection.

☒ Life sciences  ☐ Behavioural & social sciences  ☐ Ecological, evolutionary & environmental sciences

For a reference copy of the document with all sections, see nature.com/documents/nr-reporting-summary-flat.pdf

# Life sciences study design

All studies must disclose on these points even when the disclosure is negative.

| | |
|---|---|
| Sample size | The study included individuals all individuals in UK Biobank who also had measured protein levels and whole genome sequences. |
| Data exclusions | We subset to individuals of inferred European, South-Asian and African genetic ancestry using genetic principal component analysis. |
| Replication | Results were not able to be replicated due to the lack of other cohorts with whole genome sequence data and plasma protein levels measured in sufficiently large sample sizes |
| Randomization | This study is observational, so randomisation is not applicable to this type of study |
| Blinding | This study is observational, so blinding is not applicable to this type of study |

# Reporting for specific materials, systems and methods

We require information from authors about some types of materials, experimental systems and methods used in many studies. Here, indicate whether each material, system or method listed is relevant to your study. If you are not sure if a list item applies to your research, read the appropriate section before selecting a response.

## Materials & experimental systems

| n/a | Involved in the study |
|---|---|
| ☒ | Antibodies |
| ☒ | Eukaryotic cell lines |
| ☒ | Palaeontology and archaeology |
| ☒ | Animals and other organisms |
| ☒ | Clinical data |
| ☒ | Dual use research of concern |
| ☒ | Plants |

## Methods

| n/a | Involved in the study |
|---|---|
| ☒ | ChIP-seq |
| ☒ | Flow cytometry |
| ☒ | MRI-based neuroimaging |

## Plants

Seed stocks NA

Novel plant genotypes NA

Authentication NA

