## [Peer Review File · Nature Genetics]

Whole genome sequencing analysis identifies rare, large-effect non-coding variants and regulatory regions associated with circulating protein levels

Corresponding Author: Dr Gareth Hawkes

Version 0:

Decision Letter:

6th Dec 2023

Dear Gareth,

Your Article, "Whole genome sequencing analysis identifies rare, large-effect non-coding variants and regions associated with circulating protein levels" has now been seen by 3 referees. You will see from their comments copied below that while they find your work of considerable potential interest, they have raised quite substantial concerns that must be addressed. In light of these comments, we cannot accept the manuscript for publication, but would be very interested in considering a revised version that addresses these serious concerns.

In brief, the three reports are split on the overall advance presented by your study.

Reviewer #1 suggests there is some interest in the aims of the work, but their comments suggest that the advance over the Nature studies on this same data needs to be clarified.

Referee #2 has the same question of the separation between this submission and those Nature papers, but is - at this stage - more negative.

Reviewer #3, conversely, sounds supportive, and makes some useful suggestions for further analyses that would help to improve the work further.

In our reading of these reports, we think there has been sufficient support voiced to offer you the chance to improve the manuscript. We would particularly highlight the major concern of Referees #1 and #2: the distinction between your study and those at Nature must be made much clearer; we believe that these reviewers are exceedingly unlikely to be supportive of publication without a persuasive argument addressing this issue.

We hope you will find the referees' comments useful as you decide how to proceed. If you wish to submit a substantially revised manuscript, please bear in mind that we will be reluctant to approach the referees again in the absence of major revisions.

To guide the scope of the revisions, the editors discuss the referee reports in detail within the team, including with the chief editor, with a view to identifying key priorities that should be addressed in revision and sometimes overruling referee requests that are deemed beyond the scope of the current study. We hope that you will find the prioritised set of referee points to be useful when revising your study. Please do not hesitate to get in touch if you would like to discuss these issues further.

If you choose to revise your manuscript taking into account all reviewer and editor comments, please highlight all changes in the manuscript text file. At this stage we will need you to upload a copy of the manuscript in MS Word .docx or similar editable format.

*1) Include a "Response to referees" document detailing, point-by-point, how you addressed each referee comment. If no

action was taken to address a point, you must provide a compelling argument. This response will be sent back to the referees along with the revised manuscript.

*2) If you have not done so already please begin to revise your manuscript so that it conforms to our Article format instructions, available here. Refer also to any guidelines provided in this letter.

*3) Include a revised version of any required Reporting Summary: <https://www.nature.com/documents/hr-reporting-summary.pdf>
It will be available to referees (and, potentially, statisticians) to aid in their evaluation if the manuscript goes back for peer review.
A revised checklist is essential for re-review of the paper.

Please be aware of our guidelines on digital image standards.

Link Redacted

If you wish to submit a suitably revised manuscript we would hope to receive it within 6 months. If you cannot send it within this time, please let us know. We will be happy to consider your revision so long as nothing similar has been accepted for publication at Nature Genetics or published elsewhere. Should your manuscript be substantially delayed without notifying us in advance and your article is eventually published, the received date would be that of the revised, not the original, version.

Nature Genetics is committed to improving transparency in authorship. As part of our efforts in this direction, we are now requesting that all authors identified as 'corresponding author' on published papers create and link their Open Researcher and Contributor Identifier (ORCID) with their account on the Manuscript Tracking System (MTS), prior to acceptance. ORCID helps the scientific community achieve unambiguous attribution of all scholarly contributions. You can create and link your ORCID from the home page of the MTS by clicking on 'Modify my Springer Nature account'. For more information please visit please visit www.springernature.com/orcid.

Thank you for the opportunity to review your work.

Sincerely,

Michael Fletcher, PhD
Senior Editor, Nature Genetics

ORCID: 0000-0003-1589-7087

Referee expertise:

Referees #1, #2: human genetics; proteogenomics.

Referee #3: proteomics, including affinity-based.

Reviewers' Comments:

Reviewer #1:

Remarks to the Author:

The manuscript „Whole genome sequencing analysis identifies rare, large-effect non-coding variants and regions associated with circulating protein levels” by Hawkes et al is emphasizing key point in genetics, there is relevance to study non protein coding variants.

Their results are overlapping with previous reports by the 3 OLINK- UKB proteomics paper published in 2023, but the authors attempt to focus the interest on that specific point. It is not always clear if the authors are trying to conclude in a pragmatic manner on the method that can be used i.e., performing sequencing with WES vs WGS. If this is what they mean, then I suggest that they have a clearer emphasis on that.

I think the authors conclusions seem to promote the idea of WGS instead of WES. Can they take a clear position on that?
The manuscript could be shorten

Minor points:

1-page 1, title "Rare non coding variants with large effect" instead of large-effect non-coding variants

2- First sentence in abstract could be made more concise and to the point

3- Should *lair2* be *LAIR2* on Page 4

Major points:

3- The author should discuss the possibility of missing some exonic and splicing variant. The difference between WES and WGS was emphasized in Halldorsson UKB Nature 2022. Even when performing WGS some coding variants are, especially if they are complex indels, repeats, region with poor coverage. In addition, a number of new gene transcripts are added when updating maps.

4-Eldjarn et al. has performed a similar comparison of the effect of variant on PQTL direction. While doing that they made the choice of which allele to consider: Minor allele, Alternative allele or derived allele?

Can the author of the current manuscript discuss that point?

5-The 5' untranslated result should be mentioned in the abstract. Thus, the authors should recognize that these variants within 5'UTR are exonic. Interestingly Halldorsson et al. noted that Whole exome sequencing capture method used in UKB do not cover well the untranslated 5' and 3' region despite being exonic, So the term Whole exome is to take with caution

6- The sentence "Rare genetic variants in non-coding regions of the human genome can cause severe rare disease" should be moderated by adding the word "Sometime". Out of all Pathogenic and Likely Pathogenic variants in Clinvar only few are not annotated as coding or splicing.

However, since most people only sequence the coding/ splicing through WES or other gene tests, they are unlikely to cover non coding for rare disease.

7- REF 27 should move to sentence "The UK Biobank's (UKB) release of circulating protein data, in combination with WGS (REF)"

8- Are the authors using for the current paper used data from 500K release or 150K release from UKB?

9- Page 7 Missense sentence, could they also report the inverse normal transformed standardized effect (SD unit)

Have the others tried similar to Eldjarn to use the direction of the alternative allele instead of minor allele

10- Should the author discuss the potential binding artefact?

11-Should the author try to assess cisPQTL variants with eQTL?

12- Are the Fig 2 results consistent with Figure 3 in Eldjarn?

13- Page 9 when writing "We identified 777 independent rare non-coding single variant-protein associations with one of 354 proteins" can the authors compare to the number of missense and LOF

14- The following is correct: "These non-coding variants had an average absolute effect of 1.19SD (median 0.975SD), equating to 64.1% and 85.1% of the average absolute effect of rare loss of function and missense pQTLs on circulating protein levels respectively." However, the absolute effect, should be emphasizing i.e. Not taking the sign into account

15 – The mention of 999kb is misleading it is not a result it is due to the criteria. Please remind the reader of 1Mb.

16-In the sentence "Rare non-coding pQTLs were more evenly distributed between circulating protein increasing and decreasing effects (mean = -0.224 "please add the SD unit

17- Bottom of page 9, when discussing UTR, it should be noted that untranslated region is exonic but not protein coding. UTR are part of the exons of the gene which by aim should be covered by WES, but unfortunately often not see ref Halldorsson et al 2022

18- Fig 3

Please align the bars (eg. Intron splice) from the two panels. Move REG REGION to the right and expand

ALTERNATIVELY rank them by their natural location compared to gene

What do you mean by non-coding exon? Do you mean 5' and 3' or something else?

Have Y axis scaled the same

19- In the sentence: "The rare pQTL with the largest effect size was un-annotated and non-coding" what do the authors mean by un-annotated?

20- is there a copy paste error: Same MAF and P for PRELP and SIGLEC9. Also, the confidence interval for PRELP is switched

21- Page 12 when mentioning ref 24 and ASGR1

The author could add the pleiotropy is consistent with what reported by Eldjarn 2023 and also Ferkingstad 2021 (Using Somalogic)

Also, then in Nioi et al mentioned that "The del12 mutation activates a cryptic splice site, leading to a frameshift mutation" in ASGR1. It is ending up to be a LOF of an intronic origin.

22- If ASGR1 variant leading to a LOF affects 275 proteins out of this panel (and also affect some as reported by Eldjarn and Ferkingstad), it is not sure that GAS6 affect LDL. This argument is not strong

23- Page 14, the binding artefact possibility need to be discussed

24- Page 17 “Importantly, WGS data enabled us to consider more than six times the number of variants than would have been possible through single variant testing and identify associations between aggregates of rare variants and common phenotypes, using methods analogous to those used to aggregate coding variants in exome sequencing studies. “I would mention that most of these have been identified by Eldjarn and reported

25- In sentence “We show that the effect sizes of non-coding associations can have similar absolute values compared to coding associations but are more balanced between circulating protein increasing and decreasing effects. “SOMETIME as the fourth word

26-Page 17 ASGR1 sentence, mention that it is a cryptic splice in a small intron. Try to find out if it is covered in WES. The fact that it is an indel might make it difficult to call (check <https://www.science.org/doi/10.1126/science.aaf6814>)

26- Please in discussion state that the 5' UTR is exonic

Reviewer #2:

Remarks to the Author:

The present paper reports a proteogenomic analysis of the UK Biobank Olink proteomics data.

They focus is on the role of rare large-effect non-coding variants. The authors conducted a variety of analyses, that at times are hard to put into context and read more like a shopping list.

Given that three papers have been recently published in Nature on a superset of the proteomics data (3000 instead of 1500 proteins) and sample set (50,000 with imputed and/or exome sequencing data compared to 20,000 with WGS data here), the question is what discovery is truly new in this paper.

They claim that their results have important implications for the identification, and role, of rare non-coding variation associated with common human phenotypes. I do not feel that there are many important implications that have not already been raised in one way or the other in the previous three papers.

While the analyses are valid, I find most of the results rather descriptive and underwhelming.

The authors assert that some novelty resides in the rare, large-effect non-coding variants, enabled by and the use of WGS, which has not been done before.

However, I find the results coming from these analyses rather descriptive, and at places hard to put into context. A major shortcoming is that the authors almost entirely ignore challenges inherent to affinity proteomics: many of the associations are likely driven by so-called “epitope-effects”, that is, changes in binding affinity due to protein changing variants rather than genuine changes in protein levels. Rare non-coding variants may lead to allele-specific expression of protein variants with higher or lower binding affinity, which needs to be addressed.

Also, I am not convinced that the approach of conditioning rare variants combined with frequent variants is numerically stable at sample sizes as large as UKB, which may lead to spurious signals.

Regarding differences between their results and those of the previous Nature paper, the authors state:

“These differences may be partly driven by the sample size difference, and by differences in methods for conditional analysis: their analysis used forward stepwise conditional analysis to define conditionally independent pQTLs, while we performed both forward- and backward-conditional analysis steps implemented in GCTA CoJo. It is likely that these differences in methodology have led to the differences in associated pQTLs between the two studies, highlighting the difficulties of interpreting multiple independent associated variants at the same locus.”

I agree, but I do not feel that the authors have solved this problem. Taken together, I feel that this leaves this paper standing somewhat disconnected from the previous Nature papers.

Minor points / suggestions:

The definition of the Olink panels is largely a marketing decision. Enrichment analysis with respect to these panels is questionable.

Pleiotropic loci, like ASGR1, may be driven by a protein variants that interfere with the affinity assay itself, as seen for VTN1 on the SOMAscan platform.

Joint analysis of rare non-coding variants with suitable transcriptomics data could support some of the findings.

The role of blood traits as confounding factors has been largely ignored. As many of the proteins originate from blood cells, overlap of the variant associations with GWAS on blood traits might be of interest.

Reviewer #3:

Remarks to the Author:

Hawks et al. report on “Whole genome sequencing analysis identifies rare, large-effect non-coding variants and regions associated with circulating protein levels“ in which they used data from whole genome sequencing in combination with 1500 circulating proteins to determine the effect of genetic variance in non-coding regions on the abundance of the blood proteins. The study is based on recently released data from the UK Biobank Pharma Proteomic Project, citing all relevant publications. The presented work extends association studies conducted with protein-coding regions to shed some light on the influence of non-coding variants.

My expertise is in proteomics and multi-omics, so I will mainly comment on the aspects related to how the presented findings can assist in explaining the influence of non-coding variants on circulating proteins (and less so about the details regarding the methods used to identify genetic associations). Even though I am less familiar with the methods presented for testing and

annotation, their description was clear, and it felt only partly overloaded with terminologies (which are likely easier to digest for the main readership of Nature Genetics).

My main concern concerns the analysis around Fig 4 and the interpretation of these observations. The studied proteins are assembled into distinct panels mainly due to the initial sample dilution needed to bring the assay into a window with optimal performance (sensitivity and resolution). Higher abundant proteins require a higher sample dilution, whereas lower abundant proteins require assays to be performed undiluted. Labeling these panels per physiological function is, in part, branding because not all proteins included in these are strictly linked to the panels' headlines. Uncoupling the findings from these panels would make the data more technology-agnostic. Investigating the proteins using different annotations would be more insightful and informative. The authors should check for additional data in the UKB PPP publications or data hosted by Gtex (<https://www.gtexportal.org>) or the Human Protein Atlas (www.proteinatlas.org). Referring to the latter, I recommend that the authors investigate the following aspects in more detail:

The mechanism behind the presence of proteins in the circulation includes active secretion, membrane shedding, or leakage of intracellular components, see PubMed: 31772123. Focussing on secretion could provide some more direct mechanistic leads as leakage or shedding require other proteins to be involved.

The proteins' tissue of origin. As the authors stated, most abundant proteins are derived from the liver. The authors should identify which organ is the primary source using a newer reference other than the one from 1951, such as PubMed: 25613900. This will allow a more refined matching of protein expression in tissues and the tissues linked to non-coding variants. This tissue-specific view will allow more functional annotation of the regulation of protein levels by non-coding variants.

The authors list a few proteins but don't go into their functional details. For example, ASGR1 is a hepatocyte-specific receptor for glycoproteins. Can something more be said about the other proteins affected by the variant? Most affected proteins should be glycoproteins, and glycosylation is a trait of proteins undergoing active secretion. What is the consequence of the truncation? Please also investigate the most prominent proteins from each of the four panels and their pathways.

There needs to be more discussion on the functional impact of rare variants on protein levels. Can the authors speculate how decrease or increase are regulated? Is there knowledge to borrow about the effects of post-translational or post-transcriptional regulation? Would the suggested tissue and secretion analysis (or a combination thereof) reveal more insights into local (within tissue) or distant (between different tissues) regulatory processes?

Other comments:

- UKB has released new data with an additional 1500 proteins. It is recommended to include these here, as this is the only publication that provides the right suitable setting.
- Please define the black lines in Fig4
- Which proteins were included in Fig 4 (all or only 777)?
- Some p-values were 0 in Tables ST17 and 18

Version 1:

Decision Letter:

28th Aug 2024

Dear Gareth,

Your Article, "Whole genome sequencing analysis identifies rare, large-effect non-coding variants and regulatory regions associated with circulating protein levels" has now been seen by the 3 original referees.

You will see from their comments below that they appreciate the improvement made in this revision, but there are still a few important points raised. We continue to be interested in the possibility of publishing your study in Nature Genetics, but would like to consider your response to these concerns in the form of a revised manuscript before we make a final decision on publication.

Briefly, there is one new major comment from Reviewer #2 that may require additional analysis; thus we would like to see your response to these and the other remaining, primarily presentational, concerns before a final decision.

To guide the scope of the revisions, the editors discuss the referee reports in detail within the team, including with the chief editor, with a view to identifying key priorities that should be addressed in revision and sometimes overruling referee requests that are deemed beyond the scope of the current study. We hope that you will find the prioritized set of referee points to be useful when revising your study. Please do not hesitate to get in touch if you would like to discuss these issues further.

We therefore invite you to revise your manuscript taking into account all reviewer and editor comments. Please highlight all

changes in the manuscript text file. At this stage we will need you to upload a copy of the manuscript in MS Word .docx or similar editable format.

*2) If you have not done so already please begin to revise your manuscript so that it conforms to our Article format instructions, available

[here](http://www.nature.com/ng/authors/article_types/index.html).

*3) Include a revised version of any required Reporting Summary: <https://www.nature.com/documents/nr-reporting-summary.pdf>

Please be aware of our [guidelines](https://www.nature.com/nature-research/editorial-policies/image-integrity) on digital image standards.

Link Redacted

We hope to receive your revised manuscript within four to eight weeks. If you cannot send it within this time, please let us know.

Sincerely,

Michael Fletcher, PhD
Senior Editor, Nature Genetics
ORCID: 0000-0003-1589-7087

Reviewers' Comments:

Reviewer #1:

Remarks to the Author:

The authors have adequately responded to all my points.

I appreciate the update of number of probes.

Reviewer #2:

Remarks to the Author:

The authors addressed my major concern regarding the relevance of the study by extending their analysis to the full set of

WGS and proteomics data available in UKB.

As an interesting correlate they identified in the revision an important point that they believe previous papers will have missed, that is, the number of 'independent' genetic associations was associated with the sequencing coverage in the coding regions of the protein's cognate gene.

This is an important point that merits emphasis but also requires deeper characterization as to which genetic regions have been missed and for which reasons, i.e. which portion of the protein coding region was missed? How much of the missing regions are low entropy repeats?

I still find the aggregated analysis underwhelming, with p-values for the enrichment analysis on the order of 0.001 max (Fig. 5). While there may be some signal there, I am not sure how to take these associations forward. In case there is need for shorting of the manuscript, this would be the least interesting part in my view.

Reviewer #3:

Remarks to the Author:

Hawkes et al. provide a substantially revised version of "Whole genome sequencing analysis identifies rare, large-effect non-coding variants and regulatory regions associated with circulating protein levels" for which they now used 3000 instead of 1500 proteins and whole genome sequencing data from UKB. This is an appropriate and appreciated update. A key learning from this was that sequence coverage in non-coding regions influenced the reliability of the assigned associations of rare variants with protein levels.

The authors have also addressed my previous concerns and conducted their new analyses accordingly. Overall, I am OK with this version and have only minor comments remaining.

Using WGS data for pQTLs is not new and has been conducted in the following studies. Please cite and discuss their learnings using such data and sequences coverage on coding regions:

Gilly et al. <https://www.nature.com/articles/s41467-020-20079-2>

Zhong et al. <https://genomemedicine.biomedcentral.com/articles/10.1186/s13073-020-00755-0>

Thareja et al. <https://academic.oup.com/hmg/article/32/6/907/6724969>

The manuscript uses a multistep filtering process. A flowchart would be highly appreciated as it can help the reader's understanding of this procedure.

Some p-values were stated as 0 (zero) instead of values > -300 . Please recalculate these using $\log_{10}p = \log_{10}(\exp(1)) * (pt(-abs(beta/se), df=N-2, log=TRUE) + \log(2))$ - see <https://github.com/karstensuhre/tensordocker/blob/main/NOTES.md>, for example.

Version 2:

Decision Letter:

Our ref: NG-A63862R1

14th Oct 2024

Dear Dr. Hawkes,

Thank you for submitting your revised manuscript "Whole genome sequencing analysis identifies rare, large-effect non-coding variants and regulatory regions associated with circulating protein levels" (NG-A63862R1). It has now been seen by the original referees and their comments are below. The reviewers find that the paper has improved in revision, and therefore we'll be happy in principle to publish it in Nature Genetics, pending minor revisions to satisfy the referees' final requests and to comply with our editorial and formatting guidelines.

Sincerely,

Michael Fletcher, PhD
Senior Editor, Nature Genetics
ORCID: 0000-0003-1589-7087

Reviewer #2 (Remarks to the Author):

The authors responded to my remaining points

First and foremost, we would like to thank each of the reviewers for thoroughly reviewing our work, resulting in what we believe to be substantial improvement. In this revision, we have extended our analyses to a dataset including twice-as-many people (UKB 500,000 WGS release) paired with twice-as-many proteins (3,000 in total), following the release of this additional data shortly after we submitted the original manuscript. In addition, we have made a range of enhancements and adjustments according to the reviewer's comments, which we hope will be to your satisfaction.

During our revisions, we also identified a number of analytical issues we think are highly relevant to the field. First, we found that the number of 'independent' genetic associations was associated with the sequencing coverage in the coding regions of the protein's cognate gene. We believe that this is because a reduction in sequencing coverage will result in missing coding variants that could confound non-coding associations – an important point that we believe previous papers will have missed. We have thus excluded, with further justification in the manuscript, proteins where the coverage between the 5' and 3'UTRs is low (<99.5%), or where the coverage in the *cis*-window (1MBp either side of the 5/3'UTR; <99.5%).

After a suggestion from Reviewer 2, that jointly considering common and rare single variant associations may be confounded, we further found that large differences between joint and marginal effects may be another indicator of missing variants. As such, we have also excluded proteins where the maximum absolute difference between effect sizes before and after joint conditioning is greater than the 90th percentile.

As such, we believe that the accuracy of our results is much higher than in the previous submission. We do, however, also provide the association results (single variant and non-coding aggregates) for all proteins, including those which were excluded, in supplementary tables.

We believe the analytical issues we have found, the methods we have used to account for those issues, and our overall results have important implications for the previous three Nature papers on the UKB proteomic data, and for analysis of sequencing data, in particular the non-coding genome, going forward for all researchers in the field.

Reviewer #1:

Remarks to the Author:

The manuscript „Whole genome sequencing analysis identifies rare, large-effect non-coding variants and regions associated with circulating protein levels” by Hawkes et al is emphasizing key point in genetics, there is relevance to study non protein coding variants.

Their results are overlapping with previous reports by the 3 OLINK- UKB proteomics paper published in 2023, but the authors attempt to focus the interest on that specific

point. It is not always clear if the authors are trying to conclude in a pragmatic manner on the method that can be used i.e., performing sequencing with WES vs WGS. If this is what they mean, then I suggest that they have a clearer emphasis on that.

I think the authors conclusions seem to promote the idea of WGS instead of WES. Can they take a clear position on that?

The manuscript could be shorten

We thank the reviewer for this point and agree that the phenotypes studied overlap those of the previous papers. However, we would argue that the overlap between our study and the previous studies is actually quite small because we have analysed a much wider and deeper set of genetic variation and addressed a different question of importance to the wider field of common phenotype genetics: can we use aggregate based tests of non-coding rare variation as well as those in the exome and single variant tests and identify new signals? Our results indicate that we can.

We analysed nearly 200M unique million variants, which equates to 3.5x the number of variants considered in Eldjarn et. al, despite our analysis only considering the *cis* window (as compared to their whole-genome approach). Most variation in the human genome is both rare and non-coding, and this is the focus of our paper, which was not addressed in the previous papers. We have added some text to our manuscript in order to more clearly emphasise this point.

Minor points:

1-page 1, title “Rare non coding variants with large effect” instead of large-effect non-coding variants

We have changed the title to “Whole genome sequencing analysis identifies rare non-coding variants with large effects, and regulatory regions associated with circulating protein levels”

2- First sentence in abstract could be made more concise and to the point

We have shortened the first sentence slightly but wanted to try and emphasize the novelty of our work in the context of the previous challenges

3- Should *lair2* be LAIR2 on Page 4

Thank you for spotting this – it has been corrected (although the specific protein has now changed)

Major points:

3- The author should discuss the possibility of missing some exonic and splicing variant. The difference between WES and WGS was emphasized in Halldorsson UKB Nature 2022. Even when performing WGS some coding variants are, especially if they are complex indels, repeats, region with poor coverage. In addition, a number of new gene transcripts are added when updating maps.

Thank you for raising this important point. The same concern led us, given our aim was primarily to identify non-coding variants, to exclude a large number of

proteins from our primary discovery analysis due to what we considered to be poor coverage. The issue of coverage is now a major component of our work.

4-Eldjarn et al. has performed a similar comparison of the effect of variant on PQLT direction. While doing that they made the choice of which allele to consider: Minor allele, Alternative allele or derived allele?

Can the author of the current manuscript discuss that point?

We originally didn't attempt this due to the lack of overlap between our rare non-coding variants and the list of ancestral alleles covered (from biomart). We did find that there was a significant difference between the ancestrally-orientated alternate allele effect size and the rare-allele orientated effect size for rare coding variants ($P = 1.51e-21$, mean rare orientated = $-0.942SD$, mean ancestral orientated = $1.71SD$), but that the majority of our rare non-coding variants were not covered by the ancestral allele database. We've plotted the distribution of ancestral allele-orientated results below: given the lack of overlap with non-coding variants we don't believe this result adds to the novelty of the paper.

5-The 5' untranslated result should be mention in the abstract. Thus, the authors should recognize that these variants within 5'UTR are exonic. Interestingly Halldorsson et al. noted that Whole exome sequencing capture method used in UKB do not cover well the untranslated 5' and 3' region despite being exonic, So the term Whole exome is to take with caution

Thank you for the suggestion – we have added this result to the abstract. We agree that this also highlights the importance of WGS over exome sequencing – we have tried to bring this out in the discussion as well.

6- The sentence “Rare genetic variants in non-coding regions of the human genome can cause severe rare disease” should be moderated by adding the word

“Sometime”. Out of all Pathogenic and Likely Pathogenic variants in Clinvar only few are not annotated as coding or splicing. However, since most people only sequence the coding/ splicing through WES or other gene tests, they are unlikely to cover non coding for rare disease.

Thank you for the suggestion – we have added this qualifier.

7- REF 27 should move to sentence “The UK Biobank’s (UKB) release of circulating protein data, in combination with WGS (REF)”

We have moved the first instance of this reference as per your suggestion

8- Are the authors using for the current paper used data from 500K release or 150K release from UKB?

Although at the time of initial submission the 500K UKB WGS data was not released, we have re-run our analyses using the larger dataset after the additional suggestion from the other two reviewers.

9- Page 7 Missense sentence, could they also report the inverse normal transformed standardized effect (SD unit)

The effect size in SD units has been added as suggested

Have the others tried similar to Eldjarn to use the direction of the alternative allele instead of minor allele

We thank the reviewer for this suggestion but, do not believe, similarly to the analysis of ancestral alleles, which we have responded to below, that this analysis would substantially add to the publication. This is because we are primarily interested in the effect of rare variants clustered into annotation-derived regulatory aggregates.

10- Should the author discuss the potential binding artefact?

Yes you are correct: we have added the following to our discussion of limitations: “*Fifth, we were unable to take account of binding effects related to the technology used to measure protein levels. However, we did observe an enrichment of associations when limiting our analyses to the subset of 551 Gold standard proteins that had also been assayed on the Somlogic platform by DECODE, and were shown to correlate strongly with levels measured by OLINK*”.

We have also added an analysis regarding epitope effects: “*Previous studies comparing two different platforms have suggested that a high proportion of cis pQTLs could be due to epitope binding artefacts. Whilst we did not have access to the SOMALOGIC data that would enable a direct comparison, we noted that none of our 604 rare non coding variants were in at least ‘low’ LD ($r^2 \geq 0.1$) with a coding variant. We also found no evidence of a relationship between the effect size of rare non-coding variants and the maximum r^2 with a coding pQTL ($P = 0.801$), even after adjusting for variant frequency ($P = 0.763$). Our findings suggest there is little evidence of epitope effects impacting our non-coding pQTL associations*”

11-Should the author try to assess cis-pQTL variants with eQTL?

The novelty of our study is that we focused most of our analyses on the 98% of variants that are non-coding, 95% of which are rare and only present in large sequenced datasets. Hence, the majority of our variants are not available in gene expression eQTL databases that are largely limited to array based common genotypes.

12- Are the Fig 2 results consistent with Figure 3 in Eldjarn?

Our Figure 2 results are consistent with the top panel of Eldjarn's Figure 3. Our Figure shows that loss of function variants have effects which on average reduce protein levels, but there are a subset which increase protein levels, while missense variants have a more balanced effect, although on average still reduce protein levels. This is similar to that shown in Eldjarn's Figure 3.

13- Page 9 when writing "We identified 777 independent rare non-coding single variant-protein associations with one of 354 proteins" can the authors compare to the number of missense and LOF

We have added the following context to that section, including updating the numbers (note that the drop in numbers is due to the exclusion of proteins with what we consider to be low-quality *cis*-WGS data): "*We identified 604 independent rare non-coding single variant-protein associations with 369 proteins (Figs. 1&2; ST9), in comparison to rare coding variation, where we identified 985 variants annotated as high-confidence loss-of-function or missense*"

14- The following is correct: "These non-coding variants had an average absolute effect of 1.19SD (median 0.975SD), equating to 64.1% and 85.1% of the average absolute effect of rare loss of function and missense pQTLs on circulating protein levels respectively." However, the absolute effect, should be emphasizing i.e. Not taking the sign into account

Apologies for any confusion – this is exactly what we have done (we refer to absolute effect sizes in the quoted sentence).

15 – The mention of 999kb is misleading it is not a result it is due to the criteria. Please remind the reader of 1Mb.

Apologies – you are correct, we have added a caveat to this sentence.

16-In the sentence "Rare non-coding pQTLs were more evenly distributed between circulating protein increasing and decreasing effects (mean = -0.224 "please add the SD unit

Sorry for the confusion – these were the SD units. However, in the revision, we have adjusted this sentence to simply state the relative proportion which decrease protein levels in the two groups: "*Non-coding variants had an average absolute effect of 1.145SD (median 0.863SD), equating to 57.1% and 74.1% of the average absolute effect of rare loss of function and missense pQTLs on circulating protein levels respectively. Further, rare non-coding pQTLs were more evenly distributed between circulating protein increasing and decreasing effects (65.2% decreasing; Fig 2d and e), as compared to rare coding pQTLs (86.3% decreasing; P heterogeneity = 5.41x10⁻¹³).*"

17- Bottom of page 9, when discussing UTR, it should be noted that untranslated region is exonic but not protein coding. UTR are part of the exons of the gene which by aim should be covered by WES, but unfortunately often not see ref Halldorsson et al 2022

Thank you for the suggestion – we have added the following note to that section: “We note that whilst UTR variants are exonic, they are not protein coding, and we thus classified them as non-coding.” – in this context we refer to non-coding as non-protein coding.

18- Fig 3

Please align the bars (eg. Intron splice) from the two panels. Move REG REGION to the right and expand

ALTERNATIVELY rank them by their natural location compared to gene

What do you mean by non-coding exon? Do you mean 5' and 3' or something else?

Have Y axis scaled the same

Thank you for the suggestion. We have aligned the bar plots from the two panels, and ordered them according to natural location.

19- In the sentence: “The rare pQTL with the largest effect size was un-annotated and non-coding” what do the authors mean by un-annotated?

Apologies for the confusion. This was a term we used to mean variants which did not fit any of our defined annotations. We have changed this confusing term in this revision and labelled these variants as either intronic or intergenic.

20- is there a copy paste error: Same MAF and P for PRELP and SIGLEC9. Also, the confidence interval for PRELP is switched

Thank you for spotting this – we have fixed this in the revised manuscript.

21- Page 12 when mentioning ref 24 and ASGR1

The author could add the pleiotropy is consistent with what reported by Eldjarn 2023 and also Ferkingstad 2021 (Using Somalogic)

Also, then in Nioi et al mentioned that “The del12 mutation activates a cryptic splice site, leading to a frameshift mutation” in ASGR1. It is ending up to be a LOF of an intronic origin.

We have significantly shortened the section on the ASGR1 intronic deletion – we hope this is satisfactory

22- If ASGR1 variant leading to a LOF affects 275 proteins out of this panel (and also affect some as reported by Eldjarn and Ferkingstad), it is not sure that GAS6 affect LDL. This argument is not strong

We agree and have chosen to remove reference to GAS6 in the manuscript.

23- Page 14, the binding artefact possibility need to be discussed

We have exclusively reported the *joint* effects of non-coding variants, i.e. adjusted for protein-coding variation. Further, our analysis of non-coding variant aggregates was adjusted for all measured coding variants in the cognate gene.

To directly address the concerns regarding epitope effects, we have also added the following analysis to the results section: *“Previous studies comparing two different platforms have suggested that a high proportion of cis pQTLs could be due to epitope binding artefacts. Whilst we did not have access to the SOMALOGIC data that would enable a direct comparison, we noted that none of our 604 rare non coding variants were in at least ‘low’ LD ($r^2 \geq 0.1$) with a coding variant. We also found no evidence of a relationship between the effect size of rare non-coding variants and the maximum r^2 with a coding pQTL ($P = 0.801$), even after adjusting for variant frequency ($P = 0.763$). Our findings suggest there is little evidence of epitope effects impacting our non-coding pQTL associations.”*

We have additionally added a paragraph in the discussion regarding the potential impact of binding artefacts.

24- Page 17 “Importantly, WGS data enabled us to consider more than six times the number of variants than would have been possible through single variant testing and identify associations between aggregates of rare variants and common phenotypes, using methods analogous to those used to aggregate coding variants in exome sequencing studies. I would mention that most of these have been identified by Eldjarn and reported

We feel it important to stress that we differ from Eldjarn et al in two main ways:

- 1. We have performed aggregate testing of non-coding variants down to, and including, singleton variants, which were not assessed in Eldjarn et al. Indeed, our analysis includes 3.5x the number of variants considered in Eldjarn et. al, despite only considering a 2Mbp cis-window per protein**
- 2. Importantly we used a different methodology for conditional analysis compared to Eldjarn et al. These methods are important because most cis protein loci contain 2 or more signals that are likely to be in partial LD. We used a joint conditional approach, that estimates the main effects of all variants simultaneously. In comparison, Eldjarn et al used a forward-step conditioning approach which is known to produce higher ratios of false positives (e.g. PMID: 16922854).**

We have written a longer response to the points raised by Reviewer Two, with regards to value added over the three Nature papers, at the start of our responses to their comments.

25- In sentence “We show that the effect sizes of non-coding associations can have similar absolute values compared to coding associations but are more balanced between circulating protein increasing and decreasing effects. “SOMETIME as the fourth word

We have added the “sometimes’ quantifier as suggested

26-Page 17 ASGR1 sentence, mention that it is a cryptic splice in a small intron. Try to find out if it is covered in WES. The fact that it is an indel might make it difficult to call (check <https://www.science.org/doi/10.1126/science.aaf6814>)

We have added the qualifier suggested, and have considerably shortened the ASGR1 section – we hope this is satisfactory.

26- Please in discussion state that the 5' UTR is exonic
We have additionally added this qualifier to the discussion.

Reviewer #2:

Remarks to the Author:

The present paper reports a proteogenomic analysis of the UK Biobank Olink proteomics data.

They focus is on the role of rare large-effect non-coding variants. The authors conducted a variety of analyses, that at times are hard to put into context and read more like a shopping list.

Given that three papers have been recently published in Nature on a superset of the proteomics data (3000 instead of 1500 proteins) and sample set (50,000 with imputed and/or exome sequencing data compared to 20,000 with WGS data here), the question is what discovery is truly new in this paper.

They claim that their results have important implications for the identification, and role, of rare non-coding variation associated with common human phenotypes. I do not feel that there are many important implications that have not already been raised in one way or the other in the previous three papers.

While the analyses are valid, I find most of the results rather descriptive and underwhelming. The authors assert that some novelty resides in the rare, large-effect non-coding variants, enabled by and the use of WGS, which has not been done before.

However, I find the results coming from these analyses rather descriptive, and at places hard to put into context. A major shortcoming is that the authors almost entirely ignore challenges inherent to affinity proteomics: many of the associations are likely driven by so-called “epitope-effects”, that is, changes in binding affinity due to protein changing variants rather than genuine changes in protein levels. Rare non-coding variants may lead to allele-specific expression of protein variants with higher or lower binding affinity, which needs to be addressed.

Thank you for your comments. We believe that the novelty of our work is as follows:

- **The three previous papers on these protein phenotypes were limited to a maximum of 57.1 million variants in a genome-wide approach. In contrast we analyse 195 million variants in total, despite only considering an ~2Mbp *cis*-window around the protein-coding gene.**
- **We have expanded on the analyses performed in the Nature papers by performing aggregate-based association testing in non-coding regions of the genome. Our manuscript is the first to 1) identify non-coding aggregate associations – an approach that has important implications far beyond circulating protein levels, given the uncertainty about if and how we can aggregate non coding variants analogous to gene based burden testing 2) show that those associations are not driven by correlation with coding variants and 3) have enough power to characterise the nature of these associated regions. The power of this**

analysis is demonstrated by the number of non-coding aggregates which did not contain an independently significant single variant, which we have added to the abstract: *“Rare non-coding aggregate testing identified 357 conditionally independent regulatory regions ($P < 8.71 \times 10^{-9}$), after conditioning on protein-coding variants and common pQTLs. Seventy four (21%) of the non-coding aggregates were not detectable by single variant testing...”*

- By using WGS data in all ~50,000 participants we were also able to identify rare variant associations not identified by Eldjarn et al.'s imputation-based approach
- Even within the coding regions, we demonstrate that WGS can identify associations which are missed by exome-sequencing. Further, the exonic UTRs, where we demonstrate a significant enrichment of pQTLs, are poorly captured by the exome sequencing.
- We have specifically considered the impact of different annotations of rare non-coding variants. Eldjarn et. al, to our knowledge, made no attempt to stratify variants by annotation. Dhindsa et. al did perform annotation-specific analyses for coding variants, which we have shown are in agreement with our results, but also contrasted with our data showing the characteristics of associations in the non-coding genome.

We acknowledge that some of the analyses feel rather descriptive, but we have aimed to address this and highlight the impact of our findings by:

1. Our analysis now covers the full 3,000 proteins in 50,000 individuals with WGS data
2. We have minimised discussing analyses which overlap with the previous papers, instead focusing on the novel analysis of testing non-coding aggregates of variants individually too rare to analyse as single variants. Most importantly, the implications of our findings go far beyond the protein phenotypes because, before our study, there were very few examples of successful use of non-coding aggregate tests, analogous to gene burden tests, but in the 98% of the genome that is non-coding.
3. We have highlighted the value added of non-coding aggregate testing to the ability of association networks to highlight biological pathways - for example, we showed that non-coding sliding windows add above and beyond purely annotation-driven aggregates

With regards to your comment on affinity – we have attempted to address this directly by demonstrating that our results were enriched within the ‘gold standard’ proteins which the Decode paper highlighted as concordant when measured using two technologies in the Icelandic Decode cohort: *“Based on the correlation between circulating protein measures across the platforms, and the similarity of lead cis pQTLs associated with the two measures, they identified 551 (out of 2,931) proteins as highly concordant (confidence tier 1 in their ST29). Of those proteins, we found 261 presented with well-covered cis-WGS coverage. Our pQTL associations, including those involving non-coding aggregate-based tests, were enriched in these 551 proteins; despite*

representing 25.4% of the proteins we tested, they covered 31.1% (test of two proportions $P = 2.85 \times 10^{-4}$) of all our pQTLs, 32.7% ($P = 6.05 \times 10^{-5}$) of the rare pQTLs, and 37.5% of the non-coding aggregate based pQTL associations ($P = 1.29 \times 10^{-5}$).

With regards to your comments on the epitope affect (copied from our response to reviewer one): We have exclusively reported the joint effects of non-coding variants, i.e. adjusted for protein-coding variation. Further, our analysis of non-coding variant aggregates was adjusted for all measured coding variants in the cognate gene.

To directly address the concerns regarding epitope effects, we have also added the following analysis to the results section: “*Previous studies comparing two different platforms have suggested that a high proportion of cis pQTLs could be due to epitope binding artefacts. Whilst we did not have access to the SOMALOGIC data that would enable a direct comparison, we noted that none of our 604 rare non coding variants were in at least ‘low’ LD ($r^2 \geq 0.1$) with a coding variant. We also found no evidence of a relationship between the effect size of rare non-coding variants and the maximum r^2 with a coding pQTL ($P = 0.801$), even after adjusting for variant frequency ($P = 0.763$). Our findings suggest there is little evidence of epitope affects impacting our non-coding pQTL associations.*”

Also, I am not convinced that the approach of conditioning rare variants combined with frequent variants is numerically stable at sample sizes as large as UKB, which may lead to spurious signals.

Thank you for this comment. To begin with, to address your concern directly, we plotted the correlation between effect sizes of single variants before and after jointly-conditioning in GCTA-CoJo (which we apply to all $MAC \geq 5$ variants, and thus includes both rare and common variants). We saw that, in the majority of cases there is good stability before and after conditioning ($r^2=0.97$), suggesting that rare-variant conditioning is stable in these sample sizes. However, in conjunction with our findings relating to poorly-covered WGS, we found that the total number of ‘independent’ findings was correlated with maximum and average differences between marginal and joint effects in our CoJo analysis. As previously described, we believe this issue is due to missing variants, not methodological problems due to joint-conditioning. We therefore removed proteins (as described in the statement at the beginning of our responses) with differences $>90^{\text{th}}$ percentile. This results (by design) in a stronger correlation of marginal and joint effects ($r = 0.993$), as shown in the second figure below.

Comparison of Marginal and Joint Effects One (All Proteins):

Comparison of Marginal and Joint Effects Two (Subsetting Proteins based on WGS quality):

Regarding differences between their results and those of the previous Nature paper, the authors state:

“These differences may be partly driven by the sample size difference, and by differences in methods for conditional analysis: their analysis used forward stepwise conditional analysis to define conditionally independent pQTLs, while we performed both forward- and backward-conditional analysis steps implemented in GCTA CoJo. It is likely that these differences in methodology have led to the differences in associated pQTLs between the two studies, highlighting the difficulties of interpreting multiple independent associated variants at the same locus.”

I agree, but I do not feel that the authors have solved this problem. Taken together, I feel that this leaves this paper standing somewhat disconnected from the previous Nature papers.

In performing our analyses on the full set of ~50,000 individuals and ~3,000 proteins, we have refined our methodology for defining conditionally independent variants. Importantly, during the revision process, we identified an undocumented function of the GCTA-CoJo software which had an unexpected impact on our previous results. The authors of GCTA-CoJo had designed their algorithm to filter a variant from the list of independent hits if the maximum variance explained was >900x the smallest variance explained, which resulted in some rare variants being excluded from the results and

erroneously labelled as collinear. After turning that filter off, we now see much stronger correlation with the previous nature paper: *“We compared our single variant pQTL results with those of Eldjarn et al (2023)¹³, who analysed genomic data imputed from 150,119 UKB whole genome sequences in the 54,306 individuals with proteomic data. We found 2,575 of their 3,386 cis-pQTLs, (within the regions we tested; 76.1%) which directly mapped to the UKB DRAGEN WGS calls, were in strong linkage-disequilibrium ($r^2 \geq 0.8$) with at least one of our signals for the same circulating protein. The overlap was larger (848 out of 904; 93.8%) when considering only rare variants (MAF < 1%)”*

We have additionally highlighted in the manuscript that our restriction to the 1Mbp region may have also led to differences between variants identified in our analysis compared to that of Eldjarn et al. Overall, we feel that the much stronger overlap between our results, and the fact that our analyses used directly sequenced genotypes compared to the imputed of Eldjarn et. al, and the known issues with higher false-positive rates when only using forward selection (i.e. iteratively adding variants to the list of conditionally independent variants without also verifying that all variants in the list are jointly associated with the phenotype PMID: 16922854) strengthens confidence in the robustness of our results.

Minor points / suggestions:

The definition of the Olink panels is largely a marketing decision. Enrichment analysis with respect to these panels is questionable.

Thank you for the clarification on the Olink panels. Following this suggestion and our updated analysis we have now modified this analysis to remove comparison of proteins based on Olink panels, instead comparing proteins annotated as secreted or signal protein to those either annotated as membrane bound or unannotated as more biologically meaningful categorisations.

Pleiotropic loci, like ASGR1, may be driven by a protein variants that interfere with the affinity assay itself, as seen for VTN1 on the SOMAscan platform.

Thank you for your comment. We agree that this is possible – indeed the intronic deletion may be affecting a different transcript, which is not captured by Olink. We have substantially reduced our focus on ASGR1, in response to the comments that our work overlapped too much with the existing Nature papers.

Joint analysis of rare non-coding variants with suitable transcriptomics data could support some of the findings.

Thank you for the suggestion – however the novelty of our work is that we have analysed nearly 200 million variants, 95% of which are rare and have not been captured in imputed datasets or smaller sequencing datasets with extensive gene expression data. As such, large transcriptomic data combined with the rare variants we have identified from WGS data are not publicly available.

The role of blood traits as confounding factors has been largely ignored. As many of

the proteins originate from blood cells, overlap of the variant associations with GWAS on blood traits might be of interest.

Thank you for the suggestion. In the revised results, we have adjusted for fasting time at blood draw, and time since blood draw as continuous covariates, to minimise these confounding factors

Reviewer #3:

Remarks to the Author:

Hawks et al. report on “Whole genome sequencing analysis identifies rare, large-effect non-coding variants and regions associated with circulating protein levels“ in which they used data from whole genome sequencing in combination with 1500 circulating proteins to determine the effect of genetic variance in non-coding regions on the abundance of the blood proteins. The study is based on recently released data from the UK Biobank Pharma Proteomic Project, citing all relevant publications. The presented work extends association studies conducted with protein-coding regions to shed some light on the influence of non-coding variants.

My expertise is in proteomics and multi-omics, so I will mainly comment on the aspects related to how the presented findings can assist in explaining the influence of non-coding variants on circulating proteins (and less so about the details regarding the methods used to identify genetic associations). Even though I am less familiar with the methods presented for testing and annotation, their description was clear, and it felt only partly overloaded with terminologies (which are likely easier

My main concern concerns the analysis around Fig 4 and the interpretation of these observations. The studied proteins are assembled into distinct panels mainly due to the initial sample dilution needed to bring the assay into a window with optimal performance (sensitivity and resolution). Higher abundant proteins require a higher sample dilution, whereas lower abundant proteins require assays to be performed undiluted. Labeling these panels per physiological function is, in part, branding because not all proteins included in these are strictly linked to the panels' headlines. Uncoupling the findings from these panels would make the data more technology-agnostic. Investigating the proteins using different annotations would be more insightful and informative. The authors should check for additional data in the UKB PPP publications or data hosted by Gtex () or the Human Protein Atlas (www.proteinatlas.org). Referring to the latter, I recommend that the authors investigate the following aspects in more detail:

The mechanism behind the presence of proteins in the circulation includes active secretion, membrane shedding, or leakage of intracellular components, see PubMed: 31772123. Focussing on secretion could provide some more direct mechanistic leads as leakage or shedding require other proteins to be involved. (<https://www.gtexportal.org>) or the Human Protein Atlas (www.proteinatlas.org). Referring to the latter, I recommend that the authors investigate the following aspects in more detail:

The mechanism behind the presence of proteins in the circulation includes active secretion, membrane shedding, or leakage of intracellular components, see PubMed: 31772123. Focussing on secretion could provide some more direct mechanistic leads as leakage or shedding require other proteins to be involved.

We thank the reviewer for their suggestions for this analysis, and we have therefore dropped the comparisons between the Olink panels. Instead we have followed the reviewer's suggestion to examine other annotations, and compared proteins labelled as secreted or signal proteins to those labelled as only membrane bound or unannotated using the DAVID software.

The proteins' tissue of origin. As the authors stated, most abundant proteins are derived from the liver. The authors should identify which organ is the primary source using a newer reference other than the one from 1951, such as PubMed: 25613900. This will allow a more refined matching of protein expression in tissues and the tissues linked to non-coding variants. This tissue-specific view will allow more functional annotation of the regulation of protein levels by non-coding variants.

We have updated the reference as you have suggested.

The authors list a few proteins but don't go into their functional details. For example, ASGR1 is a hepatocyte-specific receptor for glycoproteins. Can something more be said about the other proteins affected by the variant? Most affected proteins should be glycoproteins, and glycosylation is a trait of proteins undergoing active secretion. What is the consequence of the truncation? Please also investigate the most prominent proteins from each of the four panels and their pathways.

Apologies if there was any confusion – we do discuss the percentage of proteins that are associated with the ASGR1 variant in a phewas: “Of the 603 proteins associated with this deletion, 600 (99.5%) were glycoproteins, representing a significant enrichment (binomial $P < 2.2 \times 10^{-16}$), in line with previous functional”. We subsequently decided against looking at the most prominent protein per-panel, based on your comments above.

There needs to be more discussion on the functional impact of rare variants on protein levels. Can the authors speculate how decrease or increase are regulated? Is there knowledge to borrow about the effects of post-translational or post-transcriptional regulation? Would the suggested tissue and secretion analysis (or a combination thereof) reveal more insights into local (within tissue) or distant (between different tissues) regulatory processes?

We have attempted to investigate this through our analyses examining enrichments among our single variants and aggregate associations compared to the background of variants/aggregates within each annotation category, which found enrichments among various annotation categories including strong enrichment in 5'UTRs and 3' UTRs in both single variants and aggregates. Within the tissue-specific analysis we also found significant enrichment among sliding-windows among blood vessel and liver specific promoters in secreted and signal proteins. To understand further how specific regions affect protein levels would require in-depth analysis of individual loci. Extending this across the full list of proteins analysed in order to draw general conclusions is beyond the scope of our analysis.

Other comments:

- UKB has released new data with an additional 1500 proteins. It is recommended to include these here, as this is the only publication that provides the right suitable setting.

As all reviewers have suggested this, we have re-performed our analyses in the 50,000 individuals with WGS on all 3,000 proteins (with the caveats described at the beginning of our responses)

- Please define the black lines in Fig4

Thank you for pointing this out – we have added this detail.

- Which proteins were included in Fig 4 (all or only 777)?

Figure 5 (the previous Fig4) contains all 1026 proteins which passed our QC criteria. We have added this information to the caption.

- Some p-values were 0 in Tables ST17 and 18

Apologies – this was due to the empirical method used to calculate p-values. We have adjusted them to properly represent the level of precision.

First and foremost, we would like to thank the reviewers for their appreciation of our responses to their comments, and for taking the time to review our substantially-altered manuscript. We have further addressed the small number of comments in the second round as follows.

Reviewer #1:

Remarks to the Author:

The authors have adequately responded to all my points.
I appreciate the update of number of probes.

Thank you for your time in reviewing our work – we are pleased that you are satisfied with the current version of the manuscript.

Reviewer #2:

Remarks to the Author:

The authors addressed my major concern regarding the relevance of the study by extending their analysis to the full set of WGS and proteomics data available in UKB.

As an interesting correlate they identified in the revision an important point that they believe previous papers will have missed, that is, the number of ‘independent’ genetic associations was associated with the sequencing coverage in the coding regions of the protein’s cognate gene.

This is an important point that merits emphasis but also requires deeper characterization as to which genetic regions have been missed and for which reasons, i.e. which portion of the protein coding region was missed? How much of the missing regions are low entropy repeats?

In our revised paper, we took a conservative approach, where we excluded analysis of proteins where either the coverage between the 5’ UTR and the 3’ UTR was below 99.5% (>99.5% of bases covered at >8 depth in >90% of individuals), or the coverage in the larger 2Mbp per-protein *cis* window was below 99.5%. While this will have filtered out proteins with very small regions of lower coverage, we were primarily concerned about being sensitive to any poorly covered regions, and due to the volume of data it was not possible to examine each protein individually to distinguish these. The number of proteins measured (~3000) means that it is beyond the scope of this work to explain all of the reasons why coverage could be low and accurately classify the coverage for each protein. We have expanded the discussion to discuss these issues, as quoted below:

“Our results suggested that the number of ‘statistically independent’ pQTL (single variant and aggregate) associations for a protein was inversely correlated with cis-genomic region sequencing coverage. Therefore, exclude likely spurious associations, unlike previous work, we have also attempted to account for imperfectly captured regions. We filtered based on a combination of coverage metrics, to compensate for incompletely captured genomic regions, and differences in regression coefficients between marginal and joint association models, to compensate for individual missing causal variants. We subsequently observed evidence for enrichment of regions previously reported to be problematic within our excluded regions, demonstrating that these issues are not unique to UK Biobank. Further analysis is required to fully understand the observed low coverage in each of the extended

cis regions, which could include low-entropy regions, copy-number variation or other reasons. Our analyses indicate how poorly captured regions can lead to potentially spurious associations, and the importance of accounting for these in WGS analyses.”

However, in order to begin addressing the issue, we did perform an additional analysis to determine whether the regions we have excluded for low-coverage are unique to the UKB dataset, or an attribute of the regions themselves. We tested for overlap between the genomic regions excluded for low coverage with a database of problematic genomic regions, generated by Genome in a Bottle and the precisionFDA Truth Challenge V2. These regions were selected for evidence of one of five problematic subsets: segmental duplications, low mappability, high/low GC-rich regions, tandem repeats and difficult-to-map X-chromosome regions. We found that the regions we excluded for low-coverage were most-strongly enriched for segmental duplications, which we have added to the results section: *“Subsequent analysis revealed that these low-coverage regions were most-strongly enriched (Methods) for genomic regions previously reported to contain segmental duplications (OR = 1.10 [1.07, 1.12], P = 1.46 x10⁻¹⁷ per additional overlapping segmental duplication; ST3), suggesting these low-coverage regions are not unique to the UKB WGS dataset.”*

We have also added details of the overlap of each protein’s 2Mbp window with these previously reported problematic regions to ST3, and an associated methods section.

I still find the aggregated analysis underwhelming, with p-values for the enrichment analysis on the order of 0.001 max (Fig. 5). While there may be some signal there, I am not sure how to take these associations forward. In case there is need for shorting of the manuscript, this would be the least interesting part in my view.

We agree with the reviewer that it is not straight forward to take forward the non-coding aggregate based signals but we believe it is important to keep in for two main reasons. 1. We believe we are breaking new ground in that few studies have performed aggregate based tests in the non-coding genome and 2. The results, including the greater enrichment in secreted vs non secreted proteins, provide further reassurance that the aggregate signals are real and not an artefacts. However, we are happy to defer to the judgement of the editor.

Reviewer #3:

Remarks to the Author:

Hawkes et al. provide a substantially revised version of “Whole genome sequencing analysis identifies rare, large-effect non-coding variants and regulatory regions associated with circulating protein levels” for which they now used 3000 instead of 1500 proteins and whole genome sequencing data from UKB. This is an appropriate and appreciated update. A key learning from this was that sequence coverage in non-coding regions influenced the reliability of the assigned associations of rare variants with protein levels.

The authors have also addressed my previous concerns and conducted their new analyses accordingly. Overall, I am OK with this version and have only minor comments remaining.

Using WGS data for pQTLs is not new and has been conducted in the following studies. Please cite and discuss their learnings using such data and sequences coverage on coding regions:

Gilly et al. <https://www.nature.com/articles/s41467-020-20079-2>

Zhong et al. <https://genomemedicine.biomedcentral.com/articles/10.1186/s13073-020-00755-0>

Thareja et al. <https://academic.oup.com/hmg/article/32/6/907/6724969>

Thank you for drawing these articles to our attention. We have added a sentence in the introduction “*Whole-genome sequencing has also been used for identification of genetic variation associated with protein levels in other studies of up to 3,000 individuals[refs .”* describing these works.

The manuscript uses a multistep filtering process. A flowchart would be highly appreciated as it can help the reader's understanding of this procedure.

We have added a flowchart describing our filtering process to the Supplementary Note, which we reference in the Methods. We have also pasted it here (see below)

Some p-values were stated as 0 (zero) instead of values > -300 . Please recalculate these using

$\log_{10}p = \log_{10}(\exp(1)) * (\text{pt}(-\text{abs}(\text{beta}/\text{se}), \text{df}=\text{N}-2, \text{log}=\text{TRUE}) + \log(2))$ - see <https://github.com/karstensuhre/tensordocker/blob/main/NOTES.md>, for example.

We have added a 'LOG10P Calculated' column where appropriate (ST2 & ST5), and noted the formula used in the Methods section. The LOG10P calculated values did not perfectly match the $-\log_{10}(p)$ calculated from the p-value output by GCTA-CoJo (where available), so we have also retained the original columns for completeness.